# Signatures of Eurasian heat waves in global Rossby wave spectra

Iana Strigunova, Richard Blender, Frank Lunkeit, and Nedjeljka Žagar

Meteorological Institute, Center for Earth System Research and Sustainability (CEN), Universität of Hamburg, Grindelberg 5, 20144 Hamburg, Germany

**Correspondence:** Iana Strigunova (iana.strigunova@uni-hamburg.de)

**Abstract.** This paper investigates systematic changes in the global atmospheric circulation statistics during Eurasian heat waves (HWs). The investigation of Rossby wave energy anomalies during HWs is based on the time series of Hough expansion coefficients representing Rossby waves with the troposphere-barotropic structures through the extended boreal summer in the ERA5, ERA-Interim, JRA-55, and MERRA reanalyses. The climatological Rossby wave energy distribution is shown to follow a $\chi^2$-distribution with skewness dependent on the zonal scale.

The applied multivariate decomposition reveals signatures of the Eurasian HWs in the probability density functions (PDFs) of the Rossby wave energy across scales. Changes in the PDFs are consistent with changes in the intramonthly variance during HWs. For the zonal mean state (the zonal wavenumber $k = 0$), a decrease in skewness is found, although not statistically significant. A reduction in skewness hints to an increase in the number of active degrees of freedom indicating more independent modes involved in the circulation. A shift in the spectral distribution of the $k = 0$ intramonthly variance is shown to describe a weakening of the mean westerlies near their core at 45°N and their strengthening at high latitudes. At planetary scales ($k = 1 - 3$), the skewness in the troposphere-barotropic Rossby wave energy significantly increases during HWs. This coincides with a reduction of intramonthly variance, in particular at $k = 3$, and persistent large-scale circulation anomalies. Based on the $\chi^2$-skewness, we estimate a reduction of the active degrees of freedom for the planetary-scale Rossby waves of about 25% compared to climatology. At synoptic scales ($k = 4 - 10$), no change in skewness is detected for the Eurasian HWs. However, synoptic waves $k = 7 - 8$ are characterised by a statistically significant increase in intramonthly variance of about 5% with respect to the climatology. In addition, a shift of the entire Rossby wave energy distribution at synoptic scales, along with amplification, is observed during HWs.

Keywords: Eurasian heat waves, Rossby waves, spectral decomposition, circulation statistics, skewness, intramonthly variance

## 1 Introduction

Heat waves, periods with the daily maximum temperatures exceeding the climatological conditions by certain thresholds, have been increasing in numbers and magnitude, especially over Eurasia (e.g., Rousi et al., 2022). While the current operational numerical weather and ensemble prediction systems forecast such extremes several weeks ahead (e.g., Emerton et al., 2022), understanding the mechanism and dynamics of heat waves poses a challenge. Heat waves (HWs) are connected with persistent high-pressure systems (blockings). Numerous studies focus on the onset and drivers of blocking; however, no consensus exists

due to complexity of dynamical and thermodynamical processes involved (e.g., Kautz et al., 2022). Blockings are often parts of large-scale quasi-stationary wave patterns (e.g., Stefanon et al., 2012). On one side, persistent weather patterns are part of internal variability. On the other side, the effect of climate change on the frequency and persistence of these patterns is still under debate (Woollings et al., 2018). For example, Park and Lee (2019) showed that these persistent weather patterns can be forced or triggered by remote anomalous tropical heating. While the physical mechanisms leading to blockings are under discussion (Petoukhov et al., 2013; Nakamura and Huang, 2018; Teng and Branstator, 2019; Wirth and Polster, 2021), the quasi-stationary behaviour of these wave patterns is shown to lead to concurrent extreme events (Kornhuber et al., 2020; Fuentes-Franco et al., 2022).

In contrast to previous studies investigating particular aspects of HWs, our research aims to identify changes in the global Rossby wave energy statistics during Eurasian HWs and to couple them to the observed circulation. While a number of studies addressed particular aspects of HWs over Eurasia (e.g., Feudale and Shukla, 2011; Schneidereit et al., 2012; Trenberth and Fasullo, 2012; Drouard and Woollings, 2018), their effects on the global spatio-temporal variance spectra have not been studied. We analyse the global three-dimensional (3D) circulation in terms of horizontal and vertical scales of the Rossby waves and compare the HWs with the climatology. As we show, the probability density function (PDF) of the Rossby wave energy, which is described by the $\chi^2$-distribution, changes during the Eurasian HWs. The changes are quantified by skewness of the PDFs for different zonal wavenumbers. The associated reduction of the number of active degrees of freedom compared to climatology can be used to explain the coarse structure of blocking events in the midlatitude troposphere.

The distributions of atmospheric fields are in general known to be non-Gaussian (Sura et al., 2005; Perron and Sura, 2013). However, the Central Limit Theorem may still be applicable when the sums of components in high-dimensional systems are involved, with assumptions of independent and identical distributions of summing components[1]. As we demonstrate, the distributions of anomalies in atmospheric energy can appear visually close to the normal distribution due to the Central Limit Theorem. However, the energy anomaly distributions are still skewed, which can be considered as an inherited property from energy ($\chi^2$-distributions). The skewness, $\gamma$, of the $\chi^2$-distribution is given by $\sqrt{8/df}$ and the excess kurtosis, $\kappa$, is given by $12/df$ with the number of independent degrees of freedom denoted $df$ (Wilks, 2011). In the $\chi^2$-distribution, the term "degrees of freedom" is defined by the number of sum of squares of independent (uncorrelated) normally distributed variables. In our analysis, the number of degrees of freedom is the number of all possible modes used in the projection, while the number of active degrees of freedom is a measure of the concentration of energy in large wavenumbers during a heat wave. It is important that localised structures like blocking do not consist of a finite set of low wavenumber modes but can also include contributions from higher wavenumbers (as is the case for Fourier series). Therefore, the number of active degrees of freedom is not a sharp condition but can be used to measure the system's complexity. Note that because the atmospheric circulation is the composite of the zonal mean state and the superposition of waves which might be dependent, the statistical properties might deviate from the ideal situation.

---

[1]Under the independence of components or variables in a high-dimensional system, one can consider that their time series are uncorrelated. The identity of distributions of summing components can be regarded in terms of their mean and variances being equal.

Advanced statistical methods are common tools in the research of extreme weather events. For example, Galfi and Lucarini (2021) analysed surface HWs using Large Deviation Theory and found that the associated persistent atmospheric patterns are not typical (in the statistical sense) compared to the climatology but follow a dynamics which is already encoded in the natural climate variability. Lucarini and Gritsun (2020) considered blockings as manifestations of Unstable Periodic Orbits and their stability as an indicator of predictability and the involved number of degrees of freedom. They find low predictability at the onset and the decay, and increased predictability in the mature phase of blocking events in the Atlantic.

A more common tool for the examination of midlatitude circulation during heat waves is the Fourier series analysis of single variable data along the latitude circles. This approach identifies anomalies in the planetary- and synoptic-scale Rossby waves during extreme events in terms of the Fourier amplitudes and phases of temperature, geopotential or wind variables at different levels. For example, Screen and Simmonds (2014) found a significant increase in the monthly variance and mean of anomalies of the Fourier amplitudes of 500 hPa geopotential heights for zonal wavenumbers 3-8 and suggested that amplified planetary waves are connected with temperature and precipitation extremes. Coumou et al. (2014) analysed wind fields at 300 and 500 hPa and found out that zonal wavenumbers $6 - 8$ are the most probable candidates for quasi-resonance (amplified quasi-stationary Rossby waves due to the resonance with free waves trapped within the waveguide) according to Petoukhov et al. (2013). More recently, Kornhuber et al. (2019) showed the coupling between the zonal wavenumber 7 in daily wind and temperature data at several standard pressure levels and surface extremes, such as HWs and floods which occurred during the boreal summer 2018.

Our heuristic approach to spectral analysis of HWs considers the horizontal and vertical scales simultaneously by using the normal-mode function (NMF) decomposition to project daily circulation fields onto Rossby and non-Rossby components (Kasahara, 2020). The NMF decomposition is multivariate meaning that the wind and geopotential variables are represented by the same spectral expansion coefficient thereby separating the circulation into the balanced (or Rossby) and unbalanced (non-Rossby) components[2].

Previous applications of the NMF decomposition showed that modal analysis complements other methods of analysing global circulation by providing scale- and dynamical regime dependent information on the variability and by quantifying it in wavenumber space (Žagar et al., 2017, 2020, 2019). Žagar et al. (2020) quantified amplitudes and trends in midlatitude traveling and quasi-stationary Rossby waves and in the equatorial wave activity in the reanalysis data. They found a statistically significant reduction of subseasonal variance in Rossby waves with zonal wavenumber $k = 6$ along with an increase in variance in wavenumbers $k = 3 - 5$ in the summer seasons of both hemispheres. However, they did not attempt to relate these changes with the surface weather or extreme events. This task is carried out in the present study.

Our goal is to investigate whether and how surface heat waves during boreal summer over Eurasia affect the global atmospheric variability spectrum. While it is not evident *a priori* that regional HWs have their signatures in the global Rossby wave spectra, we show that this is, in fact, the case. First, we demonstrate statistically significant changes in the global total energy anomalies probability density functions (PDFs) during HWs. Then, we interpret the dynamics of the planetary Rossby waves

---

[2]The real-time decomposition of the ECMWF circulation in Rossby and non-Rossby components is available on the MODES webpage https://modes.cen.uni-hamburg.de.

through the change in active degrees of freedom and in temporal variance on intramonthly scales. At last, we provide overall picture of changes in atmospheric circulation across scales.

The paper is organised as follows. The 3D decomposition method, statistical analysis and the heat waves identification algorithm are explained in Section 2. Section 3 contains results. First, we present examples of the NMF decomposition for two recent HWs. This is followed by the results of statistical analysis of spatial spectra (climatological and HWs energies) and its interpretation by filtering parts of balanced circulation back to physical space. Finally, we discuss how temporal variance spectra change during HWs. Conclusions are presented in Section 4.

## 2 Method and Data

In this section we describe our research method that makes use of the NMF decomposition and the MODES software (Žagar et al., 2015). The method is applied to the four modern reanalysis datasets. We present the criteria for Eurasian surface HWs and associated selection method for the spectral expansion coefficients.

### 2.1 Normal-mode function decomposition of global circulation

The NMF decomposition is carried out in the terrain-following, global coordinate system $(\lambda, \varphi, \sigma)$, where $\sigma = p/p_s$ is the ratio of the vertical level pressure $p$ and the surface pressure $p_s$, $\lambda$ denotes longitude and $\varphi$ is latitude. At every time step $t$, the horizontal winds $(u, v)$ and geopotential height $(h)$ on $\sigma$ levels are projected to precomputed vertical and horizontal structure functions (VSFs and HSFs, respectively). The VSFs are the numerical solutions of the vertical structure equation whereas the HSFs are eigensolutions of the Laplace equation without forcing and are given in terms of the Hough harmonics. The Hough harmonics are defined as a product of the latitude-dependent Hough functions and harmonic waves in the longitudinal direction (e.g. Kasahara, 2020). The horizontal and vertical structures are coupled by the eigenvalues of the vertical structure equation, the so-called "equivalent depth". The reader is referred to Žagar et al. (2015) and Kasahara (2020) and references therein for details of the theory.

The projection of discrete data consists of two steps. In the first step, the data vector $\mathbf{X}(\lambda, \varphi, \sigma) = (u, v, h)^{\mathrm{T}}$ is expanded into a series of orthogonal VSFs denoted $G_m$ according to

$$\mathbf{X}(\lambda, \varphi, \sigma) = \sum_{m=1}^{M} G_m(\sigma)\, \mathbf{S}_m\, \mathbf{X}_m(\lambda, \varphi)\,. \tag{1}$$

The vertical mode index $m$ ranges from 1 to $M$, the total number of vertical modes, that can be equal or less the number of vertical levels. For every $m$, the nondimensional data matrix $\mathbf{X}_m(\lambda, \varphi) = (\tilde{u}, \tilde{v}, \tilde{h})^T$ is obtained by the normalisation by the $3 \times 3$ diagonal matrix $\mathbf{S}_m$ with elements $\sqrt{gD_m}$, $\sqrt{gD_m}$, $D_m$, where $D_m$ denotes the equivalent depth of the vertical mode $m$. The nondimensional variables are denoted with $(\tilde{\ })$.

In the second step, the horizontal nondimensional motions are projected onto a series of Hough harmonics $\mathbf{H}_n^k$ for every $m$ as

$$\mathbf{X}_m(\lambda,\varphi) = \sum_{n=1}^{R} \sum_{k=-K}^{K} \chi_n^k(m)\,\mathbf{H}_n^k(\lambda,\varphi;m)\,, \tag{2}$$

where $K$ denotes the total number of zonal waves and $R$ is the total number of meridional modes. The complex Hough expansion coefficients $\chi_n^k(m)$ depend on three indices: $m$, meridional mode index $n$ and zonal wavenumber $k$. For every $n$, the projection includes two types of motions: Rossby modes[3] (quasi-geostrophic or balanced dynamics) and inertia-gravity modes that represent divergence-dominated unbalanced dynamics. The inertia-gravity modes consist of eastward- and westward-propagating solutions and together with the equatorial Kelvin and mixed Rossby-gravity waves constitute the non-Rossby modes that are not used in this study.

It is the inverse of Eq. (1) and Eq. (2) that is solved in the forward projection. The second step gives the complex Hough expansion coefficients $\chi_n^k(m)$ as

$$\chi_n^k(m) = \frac{1}{2\pi} \int_0^{2\pi} \int_{-1}^{1} \mathbf{X}_m\,[\mathbf{H}_n^k]^*\mathrm{d}\mu\,\mathrm{d}\lambda\,, \tag{3}$$

where $\mu = \sin(\varphi)$ and the asterisk (*) denotes the complex conjugate. The integrations in the zonal and meridional directions are calculated by the fast Fourier transform and the Gaussian quadrature, respectively.

MODES is applied to the four modern reanalyses: ERA5 (Hersbach et al., 2020), ERA-Interim (Dee et al., 2011), the Japanese 55-year Reanalysis JRA-55 (Kobayashi et al., 2015), and the Modern-Era Retrospective analysis for Research and Applications MERRA (Rienecker et al., 2011). We use daily data at 12 UTC from 1980-2014 (1980-2019 for ERA5) on the regular Gaussian grid that consists of $256 \times 128$ grid points in the zonal and meridional directions, respectively. Vertically the data are interpolated on the predefined 43 $\sigma$ levels. The same datasets and setup were used in Žagar et al. (2020) except that ERA5 has been extended for the period 2015-2019. The projection is carried out using the following truncations: $K = 100$, $M = 27$, and $R = 150$ which combines 50 meridional modes for the Rossby modes, for the eastward inertia-gravity and for westward inertia-gravity waves modes. Since the mixed Rossby-gravity mode is counted as the first balanced modes, the present study makes use of 49 Rossby modes for every $m$ and $k$, with the meridional mode index going from $n = 1$ to $n = 49$.

We are interested in the balanced circulation with the troposphere-barotropic vertical structure that characterises the midlatitude weather during HWs. This is taken into account by selecting a subset of the VSFs that do not change their signs within the tropopause. In the NMF decomposition, the rigid lid is at zero pressure, just like in the models used for reanalyses. The 43-level datasets extend vertically up to about 0.5 hPa so that a number of VSFs is characterised by a barotropic structure within the troposphere meaning no zero crossing below the tropopause. The first seven VSFs are shown in Fig. 1. With the middle latitude tropopause taken at 250 hPa, the VSFs with $m = 1 - 5$ can be regarded as troposphere-barotropic modes.

---

[3]We use both 'modes' and 'waves' interchangeably but the latter refers to the case without the zonal mean state ($k = 0$).

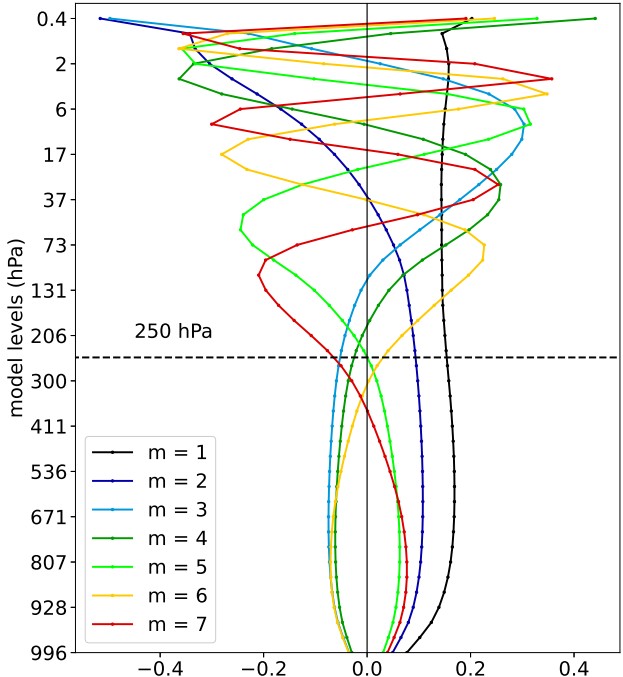

**Figure 1.** Vertical structure functions (VSFs) for the first seven vertical modes. VSFs are derived for 43 $\sigma$ levels using the stability profile of ERA-Interim data. VSFs that do not change the sign below the tropopause (defined as 250 hPa level) are troposphere-barotropic modes.

## 2.2 Heat Waves

The study area is the Eurasian region limited by the Ural mountains [35°N-65°N, 10°W-60°E]. The study area is frequently affected by HWs (e.g., Zhou and Wu, 2016), in particular, Eastern Europe and Western Russia, a location of one of the strongest HWs observed in recent decades (e.g., Barriopedro et al., 2011). For heat wave identification, we analyse daily 2 m temperature fields for the extended boreal summer (months May to September, denoted MJJAS) from 1980-2014 (until 2019 for ERA5). The identification algorithm of Ma and Franzke (2021) applies the following two criteria: (i) the temperature exceeds the 95th

percentile threshold and (ii) the duration of the exceedance is longer than three consecutive time steps (three days). Table 1 presents the list of days with HWs in the four reanalysis datasets, which is based on the algorithm. As the identification algorithm is performed independently for each reanalysis, it is expected to have discrepancies among them as seen in Table 1. Thirteen HWs are identified in ERA-Interim, JRA-55 and MERRA but the duration of HWs in individual datasets differ. Note that there are two cases with a shorter duration (2 days instead of 3 days) that were included to recognise that the four

reanalyses reproduce the same HW events. All together, there are 537 days with HWs that is about 1.5% of the total number of days, a percentage expected for extreme events.

**Table 1.** Heat waves in Eurasia during May-September 1980-2019

| | Start date | ERA5 | ERA-Interim | JRA-55 | MERRA |
|---|---|---|---|---|---|
| | | Number of detected days | | | |
| 1 | 1994-09-23 | 3 | 3 | 2 | 3 |
| 2 | 2006-06-18 | 12 | 10 | 12 | 10 |
| 3 | 2006-09-20 | 3 | 5 | 6 | 2 |
| 4 | 2007-05-20 | 12 | 12 | 12 | 12 |
| 5 | 2007-08-21 | 6 | 6 | 6 | 6 |
| 6 | 2008-09-05 | 4 | 3 | 4 | 3 |
| 7 | 2010-06-28 | 26 | 27 | 27 | 26 |
| 8 | 2010-07-27 | 21 | 21 | 19 | 21 |
| 9 | 2012-05-09 | 4 | 4 | 4 | 4 |
| 10 | 2012-06-14 | 4 | 3 | 4 | 3 |
| 11 | 2013-05-02 | 7 | 7 | 6 | 5 |
| 12 | 2014-05-17 | 5 | 3 | 3 | 3 |
| 13 | 2014-06-05 | 5 | 6 | 5 | 6 |
| 14 | 2015-06-02 | 3 | - | - | - |
| 15 | 2015-08-11 | 3 | - | - | - |
| 16 | 2015-09-17 | 11 | - | - | - |
| 17 | 2016-06-21 | 4 | - | - | - |
| 18 | 2016-08-20 | 9 | - | - | - |
| 19 | 2018-05-02 | 8 | - | - | - |
| 20 | 2018-06-27 | 4 | - | - | - |
| 21 | 2018-07-13 | 22 | - | - | - |
| 22 | 2018-08-29 | 7 | - | - | - |
| 23 | 2018-09-11 | 12 | - | - | - |
| 24 | 2019-06-01 | 3 | - | - | - |
| 25 | 2019-06-08 | 5 | - | - | - |
| 26 | 2019-06-18 | 3 | - | - | - |
| 27 | 2019-06-23 | 4 | - | - | - |
| 28 | 2019-07-24 | 3 | - | - | - |
| $\sum$days | | 213 | 110 | 110 | 104 |

## 2.3 Time series of Rossby wave energy anomalies

Our statistics makes use of Rossby wave energy anomalies during HWs in comparison to the climatology. We compute the energy time series, their anomalies and standard deviations used for standardisation, followed by combining standardised time series for all troposphere-barotropic modes and statistical analysis. In the first step, the total energy (the kinetic energy plus the available potential energy) is computed for every circulation mode $\nu$, $\nu = (k,n,m)$, as the square of the absolute value of the complex Hough expansion coefficient $\chi_\nu$:

$$I_\nu = I_n^k(m) = \frac{1}{2} g D_m \left| \chi_n^k(m) \right|^2 , \qquad (4)$$

where $g$ is the gravity. For the derivation of Eq. (4), see Kasahara (2020) or Kasahara and Puri (1981).

The time series of the daily total energy, $I_\nu(t)$, span over the MJJAS period within 35 years (1980-2014) for ERA-Interim, JRA-55, MERRA ($N_y = 35$) and 40 years (1980-2019) for ERA5 ($N_y = 40$). The climatological annual cycle is defined as an average over all years ($N_y$) for each day in MJJAS as

$$\langle I_\nu \rangle = \frac{1}{N_y} \sum_{y=1}^{N_y} I_{\nu,y} , \qquad (5)$$

and subtracted from daily energies to compute the energy deviations (or anomalies) as

$$I'_\nu = I_\nu - \langle I_\nu \rangle . \qquad (6)$$

In the further analysis, the time series of the anomalous daily energies, $I'_\nu$, is considered to be the climatological state (climatology) as a reference state for the comparison with the time series of anomalous energies during heat waves. The latter is formed combining only time steps of the observed HWs according to Table 1. For every mode $\nu$, we divide energy anomalies by their climatological standard deviation $\sigma_\nu$,

$$\tilde{I}'_\nu = \frac{I'_\nu}{\sigma_\nu} . \qquad (7)$$

The mode-wise normalisation by the standard deviation (i.e. standardisation in mathematical sense) is crucial since the energy spectrum is red not only in terms of the horizontal scales (Žagar et al., 2017), but also in terms of the vertical scale. Note that the entire time series of energy anomalies (climatology) and time series during HWs are normalised by different $\sigma$. This procedure is applied for every reanalysis independently.

The next step is to split the normalised energy anomalies of the single Rossby modes into planetary ($k = 1 - 3$) and synoptic ($k = 4 - 10$) scales, and to average over the five troposphere-barotropic modes. The mean zonal flow defined by $k = 0$ is analysed separately. For each $k$, averaging is applied also over meridional modes whenever the results are discussed in terms of the zonal wavenumber. Finally, we combine the time series of the normalised energy anomalies from the four reanalyses in the three subdomains of the global circulation: the zonal mean state, the planetary and the synoptic waves. Žagar et al. (2020) showed that the differences between climatological variance spectra for the four reanalyses are minor. Therefore, our PDFs consist of independent but similar time series. Thus, we can detect robust features of distributions of energy anomalies across different datasets.

## 3 Results

Our presentation of the results starts by showing that the selected Rossby modes from the NMF decomposition and the applied HW detection method correspond to the circulation patterns typical for the HW events. After demonstrating our methodology, we continue with the statistical analysis of the Eurasian HWs in global spectra and wrap up by coupling statistical properties with the circulation changes during HWs. But first we demonstrate in Fig. 2 that the global energy in a single Rossby mode is $\chi^2$-distributed[4]. The presented example uses the energy $I_\nu$ (Eq. 4) of the Rossby mode with $\nu = (k, n, m) = (7, 3, 1)$ which represents a part of midlatitude barotropic circulation at synoptic scales. The histogram and the fit of the $\chi^2$-distribution with two degrees of freedom, $df = 2$, correspond to the real and the imaginary parts of the time series of $I_n^k(m) = I_3^7(1)$. The Kolmogorov-Smirnov test reveals a negligible $p$-value, confirming the fit. Therefore, we find that approximation of $\chi^2$-distributed energy is satisfied to a high degree, as expected.

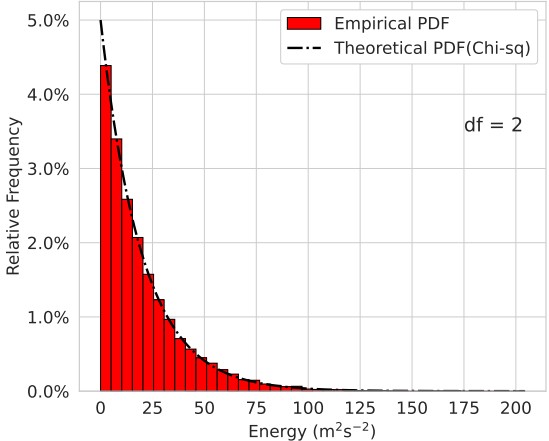

**Figure 2.** Atmospheric energy distribution for the Rossby wave with the zonal wavenumber $k = 7$, meridional mode $n = 3$ and vertical mode index $m = 1$ in ERA5 for 1980–2014. The dashed black lines correspond to the theoretical $\chi^2$-distribution ($df$ are the degrees of freedom).

### 3.1 Northern Hemisphere midlatitude circulation during heat waves

Now we demonstrate that the selected subset of vertical modes is suitable for the statistical analysis of HWs by showing the climatological state and two selected events. Figure 3a depicts the May-September balanced wave circulation (Rossby modes with $k > 0$ and all $m, n$) at $\sigma$ level close to 500 hPa. The pattern remains almost the same when only the troposphere-barotropic vertical modes, $m = 1 - 5$, are retained (Fig. 3b). This confirms our selection of the VSFs. Figure 3 is based on the ERA5 results, but other datasets provide similar results.

---

[4]The Greek letter $\chi$ used for the statistical distribution is not related to our Hough expansion coefficient $\chi_\nu$, the notation of which follows Žagar et al. (2015).

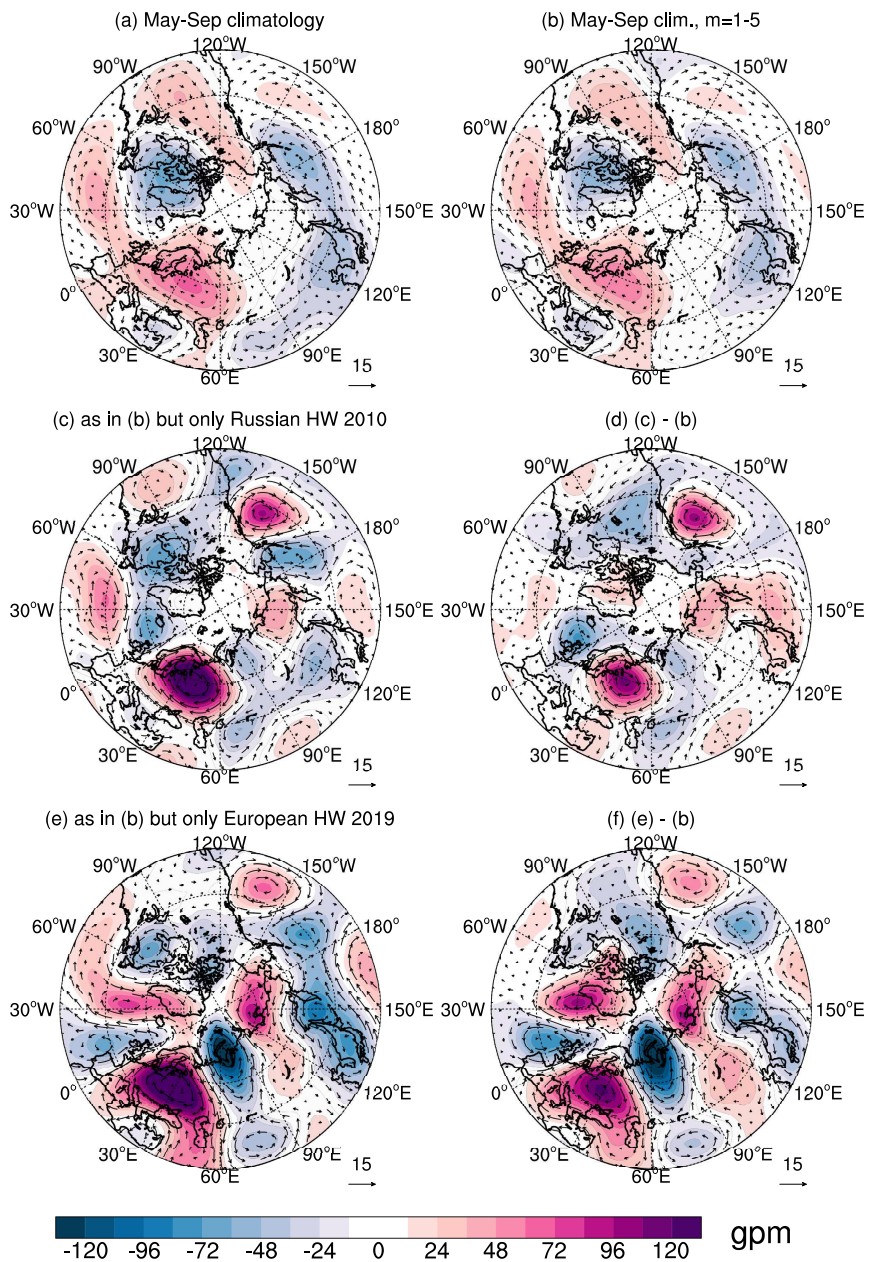

**Figure 3.** (a,b) Climatological Rossby wave circulation for extended boreal summer (MJJAS) at the $\sigma$ level close to 500 hPa in the midlatitudes. (a) All zonal wavenumbers $k > 0$, all meridional modes $n$ and all vertical modes $m$ are included. (b) As in (a), but only troposphere-barotropic vertical modes, $m = 1-5$. (c) As in (b) but for the Russian Heat Wave (HW) in 2010. (d) Difference between the (c) and (b). (e) As in (b), and (f) as in (d) but for the European HW in 2019. Coloured contours are geopotential height anomalies (in gpm). The wind speed is shown by the length of the wind vectors (with $15 \text{ m/s}^{-1}$ as a reference vector).

The circulation during the Eurasian HWs is commonly associated with the blocking and can be in the NMF-filtered circulation during two recent HW events: the Russian Heat Wave in 2010 (Barriopedro et al., 2011) shown in Fig. 3c and the European Heat Wave in 2019 (Xu et al., 2020) displayed in Fig. 3e. The difference with respect to climatology in Fig. 3b are seen in greatly enhanced amplitudes of the anticyclonic circulation over the observed surface temperature extremes (Western Russia and Europe). For the Russian Heat Wave (Fig. 3c,d), anomalies over Asia have been coupled to the Pakistan Flood (Lau and Kim, 2012). Similar, the wavy pattern along the latitudinal belt depicts teleconnections (Teng and Branstator, 2019). The difference between climatology and HWs (Fig. 3d,f) shows the meridional extension of the circulation anomalies from the tropics to the polar regions in agreement with the suggested coupling of these regions during midlatitude extremes (Behera et al., 2012). Overall, the patterns shown in Fig. 3 are qualitatively known from previous studies. The novelty is that our results are produced by multivariate filtering of the global 3D circulation allowing a scale-dependent quantification of the circulation and anomalies associated with extreme events.

## 3.2 Global statistics in Rossby-wave space: climatology

Our next step is to investigate how the Eurasian HWs affect the global spatial variability spectrum, i.e. their impact on global circulation. Here, the term global variability spectrum refers to the PDFs of the normalised anomalies in global energy, whereas the effects (or signatures) of HWs imply significant changes in the distribution of energy anomalies. The climatological PDFs are analysed for zonal wavenumbers corresponding to three ranges as described in Section 2: (i) the zonal mean state, $k = 0$, (ii) the planetary-scale circulation $k = 1 - 3$, and (iii) the synoptic-scale circulation with $k = 4 - 10$. We focus on the skewness which is not impacted by the normalisation.

Figure 4a shows the PDF for the case when all zonal wavenumbers are included in the analysis. With the skewness equal to 0.38, the PDF clearly deviates from a Gaussian distribution. A deviation from the normal distribution is found for all three ranges of wavenumbers (Fig. 4b-d). While the all ranges exhibit noticeable asymmetry, the skewness for the zonal mean and planetary-scale wave PDFs is almost twice greater than the one of the synoptic-scale waves. In addition, we note that the distributions for the zonal mean state and the planetary scales are broader than for the synoptic scales. This may reflect more time scales with a larger range of magnitudes being associated with large-scale variability in comparison to the synoptic scales.

Focusing on the skewness and kurtosis of the PDFs, Figure 5 shows box-plots of the respective parameters for the all four PDFs. Both the climatology and the HWs are presented in the figure, but the latter will be discussed in the next section together with the HW PDFs. The robustness of the statistical analysis is checked by applying bootstrapping with replacement for skewness and excess kurtosis with 1000 realizations for every presented wavenumber range. All results are found to be within the defined 95% confidence intervals (CI) for each wavenumber range (not shown). The skewness and kurtosis show that the normalised energy anomaly distribution has the highest asymmetry at the planetary scales and the zonal mean circulation seen as extended right tails in the PDFs in Fig. 4. The different numbers of contributing modes can partly explain the different skewnesses in the four wavenumber ranges. However, changes in the dynamics, such as during HWs, can modify the skewness and the active degrees of freedom, as discussed next.

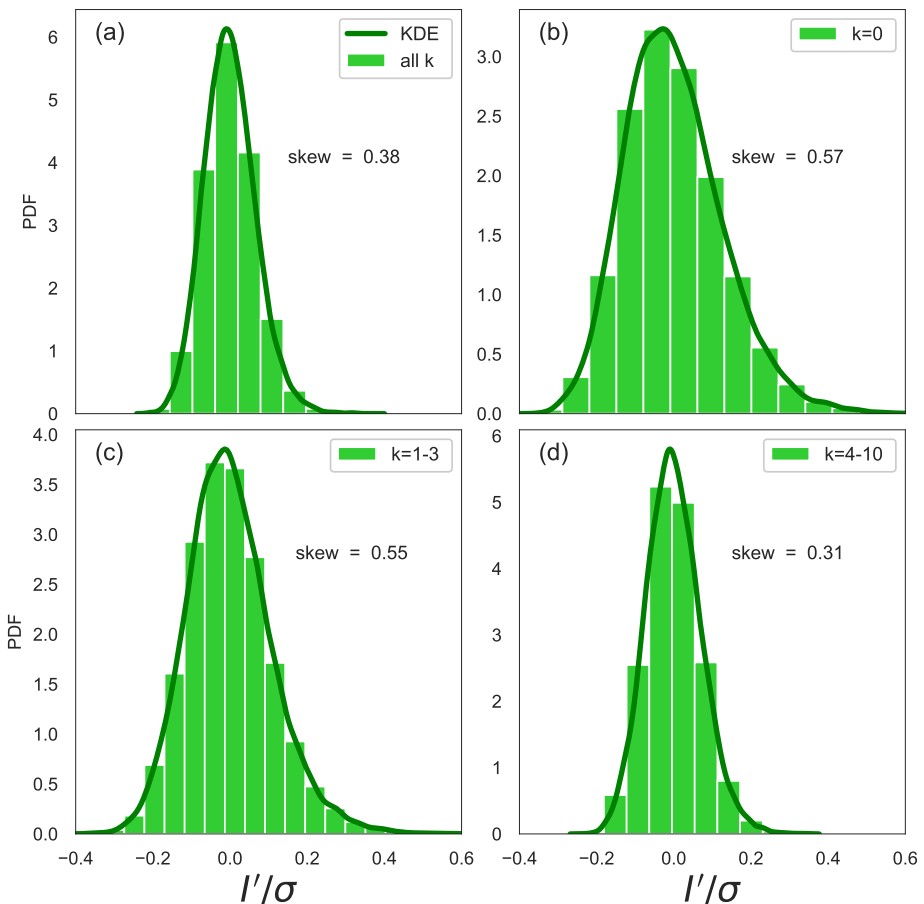

**Figure 4.** PDFs of the normalised energy anomalies in the global balanced (Rossby mode) circulation for (a) all wavenumbers $k$, (b) the zonal mean state ($k = 0$), (c) planetary-scale waves ($k = 1 - 3$), (d) synoptic-scale waves ($k = 4 - 10$). The empirical PDFs are depicted as green bars. The dark green curve is the Kernel Density Estimator (KDE).

### 3.2.1 Changes in the Rossby-wave energy statistics during heat waves

Now we compare PDFs during the observed HWs over Eurasia with the climatology in terms of the skewness and excess kurtosis that diagnose the changes in shape, especially in the tails of distributions.

The PDFs of the normalised energy anomalies in Fig. 6 demonstrate how probabilities of the energy deviations change during HWs. For the normalised total energy anomalies (all $k$; Fig. 6a) the PDF becomes broader with a longer positive tail

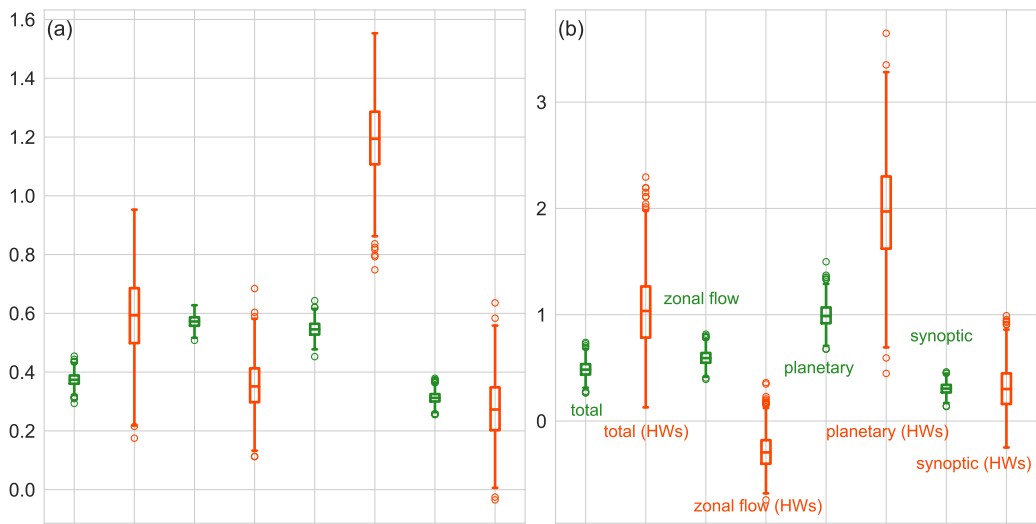

**Figure 5.** Box-plots for the (a) skewness and (b) excess kurtosis of the PDFs of normalised energy anomalies for four circulation components: all Rossby modes (all $k$), the zonal mean flow ($k = 0$), planetary-scale Rossby waves ($k = 1 - 3$), and synoptic-scale Rossby waves ($k = 4 - 10$). Vertical lines mark 95%-confidence intervals. Green and red shades denote the climatology and HWs, respectively.

indicating more high energy extremes. For the zonal mean flow ($k = 0$; Fig. 6b) only small changes are visible on the first view. The PDF of the planetary waves ($k = 1 - 3$; Fig. 6c) shows a shift of the maximum towards negative values and more positive values. While the aforementioned changes of the entire PDFs are not significant, we identify a statistically significant
change (according to the Mann-Whitney U test with 95%-confidence) in the PDFs of synoptic Rossby waves ($k = 4 - 10$; Fig. 6d). Here, the complete distribution is shifted to higher values without change in its shape. The shift can be interpreted as increased positive deviations in the synoptic-scale energy during HWs. More energy in synoptic-scale circulation can be viewed as more intensive cyclones and anticyclones which are found to maintain blocking by eddy straining (Shutts, 1983) and selective absorption (Yamazaki and Itoh, 2013) mechanisms.

How do the skewness and the excess kurtosis change during the Eurasian HWs? An increase (decrease) in skewness hints to less (more) active degrees of freedom, which can be interpreted as less (more) independent modes contributing to the variability. This can be caused by both a change in the number of contributing modes and a change of temporal coherence between different modes contributing. An increase of excess kurtosis reflects a rise in the probability of extreme values.

Together with the climatology, Figure 5 shows the skewness and the excess kurtosis for the HWs. For HW events, the two
quantities change in qualitatively the same way for different ranges of the wavenumbers. While we find almost no changes for the synoptic waves, changes are the largest at the planetary scales in the excess kurtosis (Fig. 5b). In this case, the excess kurtosis for extreme events is approximately twice larger than climatology, which reflects a rise in the probability of extreme values. The opposite change is found for the zonal mean flow, where skewness and the excess kurtosis decrease; this implies that the distribution has less extreme values. We conclude that anomalies of the planetary-scale circulation show relatively

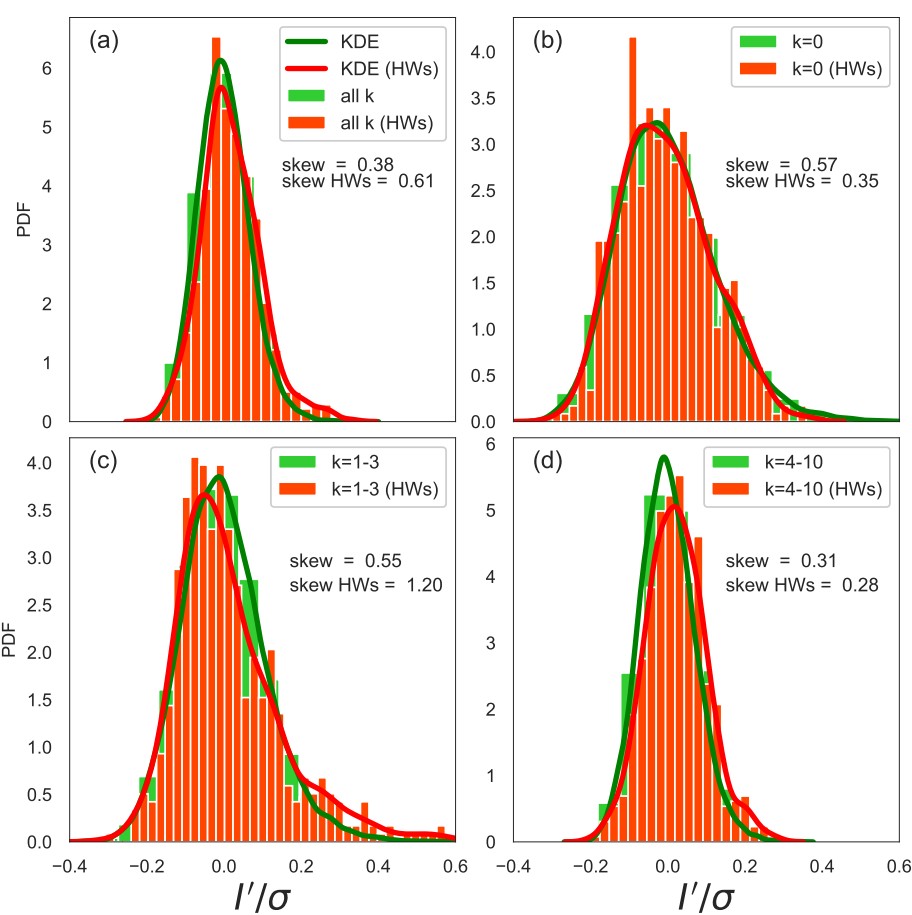

**Figure 6.** As in Fig. 4, but for the Eurasian heat waves listed in Table 1.

less (and more coherent) variability in general and persistent anomalies are generated as shown in Fig. 3d, f, although positive extremes are more likely. On the other hand, the zonal mean flow anomalies become in general weaker in agreement with Coumou et al. (2014).

The change in skewness allows for the estimation of the change in the active degrees of freedom during the HWs compared to the climatology. For the estimation, we use the exact relation for the skewness of $\chi^2$-distributed variable, $\gamma = \sqrt{8/df}$, where $df$ is the number of squares of the independent Gaussian variables with a unit variance which defines the $\chi^2$-distributed variable. We use $df_e/df_c = \gamma_c^2/\gamma_e^2$, which says that the ratio between the number of active degrees of freedom during HWs and climatology, $df_e$ and $df_c$ respectively, is equal to the ratio of their skewnesses $\gamma_e$ and $\gamma_c$, respectively. For the planetary waves which shows the largest change, the estimated $\gamma_e \approx 1.2$ and $\gamma_c \approx 0.6$ (see Fig. 5a) yield a reduction of the active degrees of freedom of about $25\%$ during HWs.

Finally, we make a note on the fact that the changes in PDFs during the Eurasian HWs apply to the global atmosphere. Our Rossby modes consist of symmetrical ($n$ odd) and asymmetrical ($n$ even) components with symmetry with respect to the equator defined for the geopotential height and zonal wind fields. We checked that both symmetrical and asymmetrical parts contribute to the PDFs of all meridional modes. In other words, the Rossby waves in the Southern Hemisphere might have contributed to the results presented. However, taking into account the lower frequency of atmospheric blocking (Wiedenmann et al., 2002) in the Southern Hemisphere, we may assume that this influence is negligible.

### 3.3 Changes in planetary-scale circulation during heat waves

The changes in the PDFs for different scales can be physically interpreted by filtering selected Rossby waves to physical space, similar to what has been done in Fig. 3. Instead of case studies, now we present the planetary-scale circulation averaged over all days with observed extremes. As earlier, we show the horizontal circulation at ERA5 $\sigma$ level near 500 hPa as representative for the troposphere-barotropic circulation.

Figure 7a is very similar to Fig. 3b which included all zonal wavenumbers. Figures 7b,c reveal that during the Eurasian HWs, a large enhancement of the positive geopotential height anomaly over Northern Europe and a negative geopotential anomaly over the North Atlantic and central Asia takes place. The vertical cross sections along the latitude circle $54°$N reveals the expected troposphere-barotropic vertical structure of anomalous circulation that extends throughout the lower stratosphere (Fig. 7d,e). The northward winds over Europe ($0° - 30°$E) and southward winds over the Asian part of Russia ($60° - 90°$E) are enhanced during HWs. Overall, we find an increase in wave amplitudes, and change in phases as can be noticed by west- and northward shifts in Fig. 7b,c and Fig. 7d,e in the Baikal lake area ($90° - 120°$E). The results in Fig. 7 align with Teng and Branstator (2012) and Ragone and Bouchet (2021), where the zonal wavenumber $k = 3$ pattern was found dominant for HWs that occurred in the US, France and Scandinavia. Therefore, the results demonstrate that changes in atmospheric circulation during surface extremes occur not only regionally but also in remote regions, similar to the idea of teleconnection patterns noted in recent studies (e.g., Kornhuber et al., 2019).

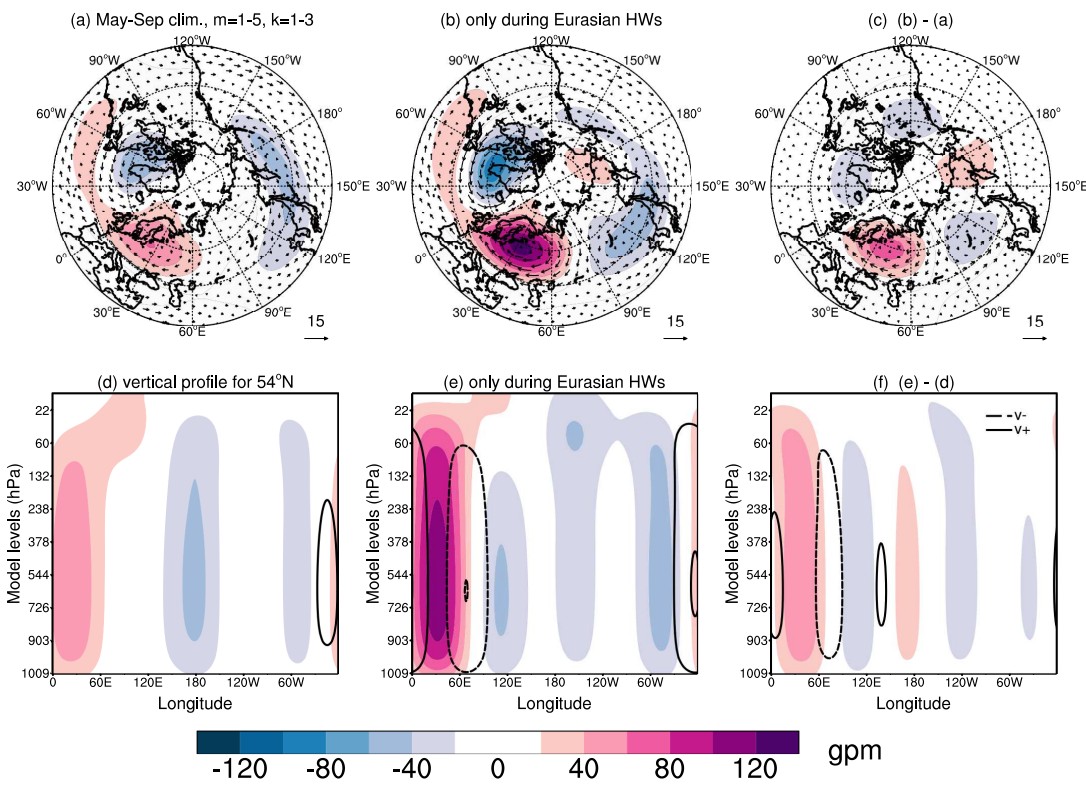

**Figure 7.** Planetary-scale, troposphere-barotropic Rossby waves ($k = 1 - 3$, $m = 1 - 5$, all $n$) at the $\sigma$ level close to 500 hPa in ERA5. (a) Mean circulation in May-September in 1980-2019, (b) composite of 28 Eurasian Heat Waves (HWs) presented in Table 1, (c) difference between (b) and (a). Coloured contours are geopotential height anomalies, every 20 gpm. The wind speed is shown by the arrow length. (d)-(f) Longitude-pressure cross sections of planetary-scale geopotential height (colours) and meridional wind (isolines) perturbations along 54°N. (d) Climatology, (e) HWs, (f) difference between (d) and (e). Solid and dashed contours in (d)-(f) correspond to the northward and southward meridional wind speed, respectively, every 2 ms$^{-1}$.

## 3.4 Changes in intramonthly variance during the surface heat waves

So far, we discussed signatures of HWs in spatial variance (energy). Now we investigate related changes in temporal variance on intramonthly scales. The temporal variance and its square root, variability, are usually studied at single points or using the time series of atmospheric indices such as North Atlantic Oscillation. The global intraseasonal variance was analysed by Žagar et al. (2020) who showed statistically significant trends in both midlatitude Rossby waves and in large-scale equatorial waves. Here, we compare the climatological intramonthly variance with that for the months with the observed Eurasian HWs in all reanalyses.

The unbiased variance $(\mathrm{Jkg^{-1}})$ is computed as

$$V_\nu = \frac{1}{N-1}\sum_{t=1}^{N} gD_m|\chi_\nu(t) - \overline{\chi}_\nu|^2, \tag{8}$$

where $\overline{\chi}_\nu$ is the monthly mean and $N$ is the number of days in a single month. As the 3D NMF expansion is a complete representation of the system, the components $\nu$ of the state vector are statistically independent and correspond to independent degrees of freedom, as discussed in Section 2. The zonal wavenumber variance spectra are obtained by summing the variances in the five vertical and all meridional modes as previously. Intramonthly variance is computed for all months and averaged to create the climatological variance spectrum, $V_\nu$. The averaging over all months with heat waves gives us the HW variance spectrum $V_\nu^h$ (here we drop extra signs for the averaging operator). The relative change in intramonthly variance due to HWs is

$$\frac{V_\nu^h - V_\nu}{V_\nu} \quad \text{or} \quad \frac{V_\nu^h}{V_\nu} - 1. \tag{9}$$

The global intramonthly Rossby wave variance spectrum is shown in Fig. 8a. It is a red spectrum, similar to the subseasonal variance spectra in Žagar et al. (2020). The redness of the spectra in Fig. 8a makes differences between the climatology and HWs difficult to detect, but they are made clear by zooming in the planetary and synoptic scales displayed as an inset panel. It shows a variance reduction of about 6% in the zonal wavenumber $k = 3$ along with the 5% variance increase in $k = 7-8$. We note that the changes in intramonthly variance are consistent with the shifts of the maxima of the respective normalised energy anomaly PDFs (Fig. 6c,d). In addition, the reduction of planetary wave intramonthly variance is also consistent with the appearance of the persistent large-scale anomaly shown in Fig. 7.

The blue shading around the variance spectra in Fig. 8a depicts the 95%-CI obtained through bootstrapping. It suggests the largest uncertainty at planetary wavenumbers. The variance reduction at $k = 3$ is within 95%-CI and, therefore, is insignificant. At $k = 7-8$, the intramonthly variance during HWs is slightly outside of the CI; therefore, the variance change is considered significant. We note here that our findings are based on a relatively small sample of identified HWs and that many events lasted under a week. To provide stronger evidence, GCM simulations can be performed, which is the scope of the future studies.

A more detailed view of the changes in the global intramonthly variance during HWs is provided in Fig. 8b including also the mean zonal state. The variance reduction at $k = 3$ and an increase at $k = 7-8$ are seen across multiple meridional modes $n$ in agreement with the midlatitude character of HWs. The quantitatively largest variance change is however seen in the zonal

mean state $k = 0$ with a positive and negative change in the two asymmetrical meridional modes, $n = 4$ and $n = 6$, respectively. The change in $k = 0$ can be explained using the latitudinal profile of the zonal-mean zonal wind presented in Fig. 9. First, it shows that the maximum zonal-mean zonal wind at 45°N during HWs is about 10% weaker than the climatology and slightly shifted (about 1°) northward. The jet near 45°N is more confined in the troposphere, with the $10~\mathrm{ms}^{-1}$ isoline near $300~\mathrm{hPa}$ compared to $200~\mathrm{hPa}$ in climatology. This means that the vertical shear of the mean zonal wind decreases during the Eurasian HWs.

Another features of the HWs seen in Fig. 9 are twice stronger zonal-mean zonal winds in the latitude belt between 60°N and 90°N with a peak difference of up to $3~\mathrm{ms}^{-1}$ at 75°N. The dipole shape of the difference in Fig. 9c is in Fig. 8b seen as a variance decrease in the meridional mode $n = 4$ and an increase in $n = 6$. Note that Fig. 9 is obtained by filtering $\overline{\chi}_n^0$ to physical space. Similar filtering for any horizontal or vertical scale of interest is straightforward, which makes the holistic modal-space statistics an attractive global complement to the single-variable, single-level Fourier analysis. We speculate that the enhancement of high latitude $k = 0$ zonal winds is a component of more-persistent double jets over Eurasia during HWs recently discussed by Rousi et al. (2022).

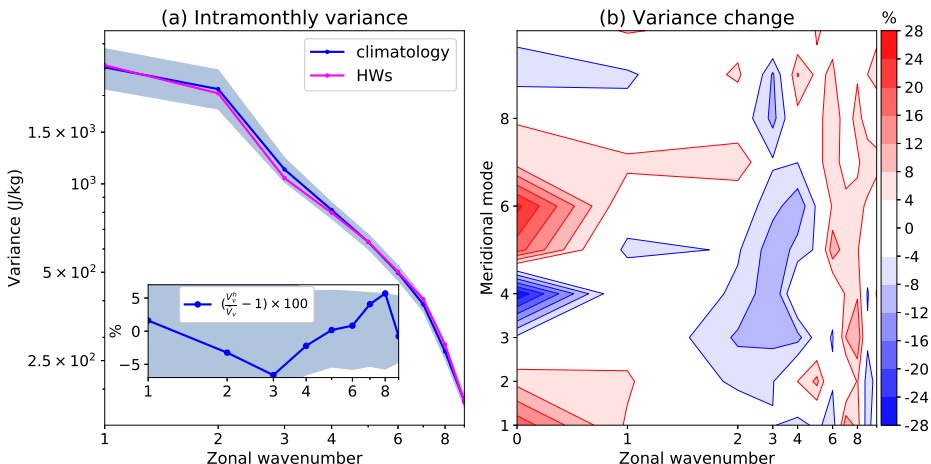

**Figure 8.** (a) Intramonthly variance spectra of the Rossby waves for the climatology (blue) and Eurasian heat waves (magenta). The embedded figure shows the relative change in percentages of the climatology. The blue shading denotes the 95%-confidence intervals. (b) Changes in the intramonthly variance with respect to climatology as a function of the zonal wavenumber and meridional mode including the zonal mean state.

## 4   Conclusions

Extreme events such as surface HWs are accompanied by changes in atmospheric circulation across many scales. Our study shows that Eurasian HWs have signatures in the global balanced circulation. The changes in global statistics of the Rossby-wave variance are made evident by analysing the four modern reanalyses: the ERA5, ERA-Interim, JRA-55, and MERRA

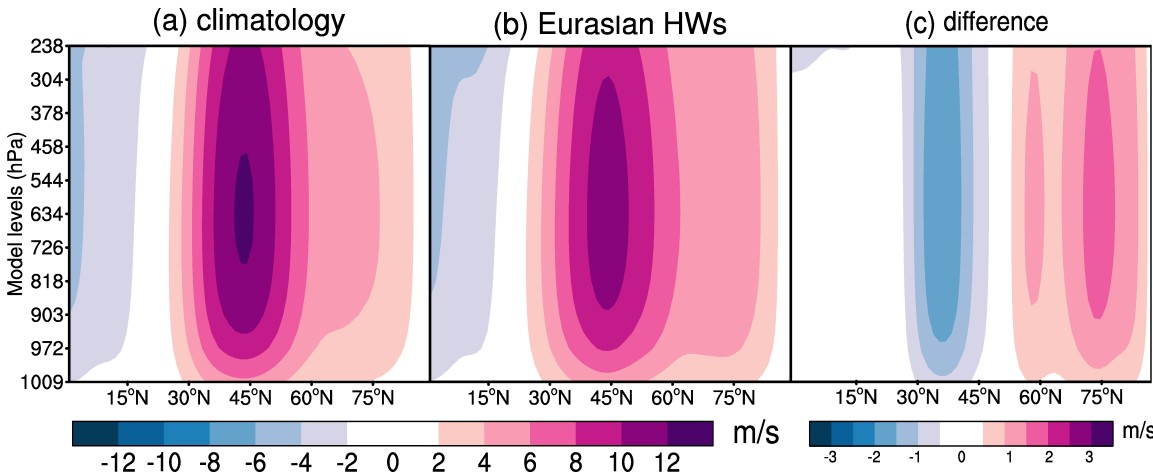

**Figure 9.** Zonal-mean zonal wind in the northern hemisphere troposphere in 1980-2019, May-Sep ERA5 data. (a) Climatology, (b) Eurasian heat waves (HWs) and (c) climatology - HWs.

datasets. The Rossby waves are identified by a multivariate projection of the global horizontal winds and geopotential height on the eigensolutions of the linearized primitive equations on the sphere with a basic state at rest (the so-called normal-mode functions). A complete projection basis provides global statistics of Rossby waves as a function of the zonal wavenumber, the meridional mode index and the vertical modes associated with the vertical structure functions. The method includes scale-350 selective multivariate Rossby-wave filtering in physical space offering an attractive global complement to the single-variable, single-level Fourier analysis.

Our analysis focuses on the Rossby waves with the barotropic structure within the troposphere that is characteristic of the midlatitude circulation during HWs. The reconstructed physical space picture of the Eurasian HWs is in agreement with previous studies (Lau and Kim, 2012; Behera et al., 2012; Coumou et al., 2014; Teng and Branstator, 2019). We find largely 355 increased amplitudes of the positive geopotential height anomaly over Northern Europe, otherwise typical for the extended summer period, and a negative geopotential anomaly over the North Atlantic and central Asia. The anomalous circulation extends throughout the lower stratosphere. In addition, there are westward and northward shifts in the circulation. During HWs, the zonal mean westerlies somewhat weaken near their climatological maximum at 45°N but get twice stronger in high midlatitudes (centred at 75°N). Future work should couple these findings with the study by Wirth and Polster (2021) on the 360 role of Rossby waves in processes leading to the double jet formation, recently discussed for Eurasian HWs by Rousi et al. (2022).

The statistical analysis is carried out on the complex time series of the Hough expansion coefficients representing Rossby modes across many horizontal scales with the troposphere-barotropic vertical structure. We demonstrate that the energy distri-

bution of a single mode follows a $\chi^2$-distribution. Statistics of the normalised energy anomalies shows that the zonal mean state $(k = 0)$ and the planetary-scale $(k = 1 - 3)$ circulation are more skewed than the synoptic and smaller scales, with extended right tails. Increased skewness of the distribution hints to the reduction in active degrees of freedom. This can be interpreted as less independent modes contributing to the observed variability, either because the number of total modes is smaller or because there is temporal coherence between different modes.

During the Eurasian HWs, the skewness in planetary-scale Rossby waves increases while the opposite occurs in the zonal mean state. The increase in skewness for planetary-scale waves reveals the decrease in the number of active degrees of freedom during HWs. This aligns with the results of Lucarini and Gritsun (2020) which are based on the atmospheric stability during Atlantic blockings. Based on the $\chi^2$-skewness, we estimate a reduction of the active degrees of freedom for the planetary-scale Rossby waves during Eurasian HWs of about 25% compared to climatology.

Consistent changes in wavenumber space are found in the intramonthly variance. Eurasian HWs are characterised by a statistically significant increase of about 5% in the intramonthly variance at synoptic scales $k = 7 - 8$, with respect to climatology. This is consistent with increased synoptic activity during blocking (e.g., Shutts, 1983; Yamazaki and Itoh, 2013). In contrast, a reduction of intramonthly variance in $k = 3$ of about 6% is found not to be statistically significant. Future studies with longer datasets, such as climate model outputs are an opportunity for models' validation and larger datasets of extreme events.

Despite the uncertainties due to the limited sample size, our results provide the following overall picture consistent with previous studies. During HWs, the planetary-scale Rossby waves (primarily $k = 3$) exhibit reduced intramonthly variability. The involved modes are less independent from one another, and a persistent large-scale anomaly is formed, typically referred to as blocking. On the other hand, the intramonthly variability of the synoptic Rossby waves increases, particularly at the zonal wavenumbers $k = 7 - 8$. The contributions of more active meridional modes to the zonal mean flow during HWs, perhaps excited by eddy-mean flow interactions, shows as an enhancement of the mean westerlies north of 60°N and their weakening near 45°N.

*Code and data availability.* The ERA-Interim and ERA5 reanalysis datasets are available via http://www.ecmwf.int. The MERRA and JRA-55 are available at https://gmao.gsfc.nasa.gov/reanalysis/MERRA and https://jra.kishou.go.jp/JRA-55, respectively. The MODES software can be requested via https://modes.cen.uni-hamburg.de. The time series of the Hough expansion coefficients for the four reanalyses are available upon the request from the authors.

*Author contributions.* All authors contributed to the study conception and design. IS developed the algorithm, performed the data analysis and wrote a first draft of the manuscript. All authors participated in data interpretation and revised previous versions of the manuscript. All authors read and approved the final manuscript.

*Competing interests.* The authors declare that the research was conducted in the absence of any commercial or financial relationships that could be construed as a potential conflict of interest.

*Acknowledgements.* This work was funded by the Deutsche Forschungsgemeinschaft (DFG, German Research Foundation) under Germany's Excellence Strategy – EXC 2037 'CLICCS - Climate, Climatic Change, and Society' (CLICCS, A6) – Project Number: 390683824, contribution to the Center for Earth System Research and Sustainability (CEN) of Universität Hamburg. We thank the former MODES group members at the University of Ljubljana Damjan Jelić and Khalil Karami for the MODES decomposition of the four reanalysis data, to Žiga Zaplotnik for his advice on processing the Hough coefficients in Python, to Qiyun Ma for the algorithm for the heat wave identification and 400 to Frank Sielmann for the variance analysis and technical support. We would also like to thank Valerio Lucarini for the discussion, and two anonymous reviewers and the Editor Dr. Gwendal Rivière for their constructive comments on the manuscript.

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
