# Peer review of "Signatures of Eurasian heat waves in global Rossby wave spectra"

_Weather and Climate Dynamics, 2022_

## Author Comment (AC1)

Paper: wcd-2022-23, entitled "Signatures of midlatitude heat waves in global Rossby wave spectra",

By Iana Strigunova, Richard Blender, Frank Lunkeit, and Nedjeljka Žagar

**Response to the comments by Referee RC1**

https://doi.org/10.5194/wcd-2022-23-RC1

Dear Referee,

Thank you very much for the comments on our manuscript.

Following your comments and comments from another reviewer and the editor, we have been largely rewriting the manuscript in an effort to highlight the original aspects of our method and the originality and added value of our results. While the results of statistics remain unaltered, we plan to extend the analysis by showing the zonal- and meridional-scale dependent entropy reduction during the Eurasian heat waves in relation to intramonthly variance reduction. For this purpose, the results section is being extended and some new figures will be added. We also plan to replace 'midlatitude' by 'Eurasian' in the title, as a more correct wording for the paper content.

Enclosed please find our response, presented in blue font follow your comments in black font.

Your sincerely,

Iana Strigunova, Richard Blender, Frank Lunkeit, and Nedjeljka Žagar

**Comment:**
**Summary of the article**
This manuscript isolates heat waves events, defined as episodes which spatially averaged surface temperature in Eurasia are above 95% percentile on at least 3 consecutive days, and compare their spatial spectra of energy anomalies of various ranges of wavenumbers with the climatology. This work finds that during heat waves, the skewness of planetary waves grows while that of the zonal mean flow goes in opposite directions, which is consistent with previous findings.
**General comments**
I am concerned about the fact that the analyses done in this work all confirm previous findings but provide no additional insights to how we understand heat waves/blocking. It has been well-known that heat waves are associated with blocking since blocking, by definition, refers to the phenomena in which (1) the eastward zonal wind is disrupted (increase in wave amplitude essentially implies weakened zonal wind, as implied by non-acceleration theorem) (2) the high pressure cell remains stagnant for a few days, (Therefore, reversal of zonal wind

and geopotential height anomalies have been used to define/detect blocking.) which, this manuscript essentially shows.

There have been many studies showing blocking being the culprit of heat waves in the eurasia region as confirmed by the analyses here. Therefore, in order to be accepted for publication, the authors should have shown what extra insights about heat waves can we obtain from these analyses, which I think the current version of manuscript is lacking.

**Response**:

We thank the Reviewer for the comments expressing a concern about the lack of added insights about heat waves in our analysis. We think that our efforts to validate the use of three-dimensional, global spectral decomposition in terms of normal-mode functions as a suitable tool for signatures of regional heat waves in global circulation, somehow masked the originality and added value of our results. While we shall re-write parts of paper to point out original aspects of our study, let us state that to our knowledge, previous studies did not quantify the skewness of global circulation at different scales of the Rossby waves during the Eurasian heat waves. We are neither aware of other research showing what aspects of the global circulation, as measured by PDFs of anomalies in the total energy, are affected by the Eurasian heat waves. Novelties of our study can be split in three parts:

1. **Method:** Our method is novel in its identification of Rossby waves using a multivariate spherical projection on the Rossby eigensolutions of the linearized primitive equations. This provides time series of Rossby waves in terms of Hough harmonics, that can be seen as the spherical equivalents of the geostrophic stream function on the midlatitude beta plane. Scale-selective Rossby wave filtering in physical space is seen as an advantage compared to univariate filtering using the Fourier series along the latitude circles. While we focus on Rossby waves or balanced dynamics, the difference to the total signal in terms of inertia-gravity modes can be used to analyse whether midlatitude ageostrophic flow (unbalanced dynamics) increases during the Eurasian heat waves. We combined the four modern reanalysis datasets to provide robust results.

2. **Statistical properties of global balanced circulation during Eurasian heat waves:** We show that the energy distribution of a single Rossby mode follows a $Chi^2$-distribution. Our scale-dependent statistics, performed on the normalized energy anomalies, shows that the energy distributions of the zonal mean state (zonal wavenumber k=0) and of the planetary-scale (k=1-3) circulation are more skewed than the distributions at synoptic and smaller scales, with extended right tails. During the Eurasian heat waves, the skewness in planetary waves increases while the opposite occurs in the zonal mean flow. The increase in skewness can be linked with a decrease in the number of active degrees of freedom in state space during heat waves. This aligns with the results of Lucarini and Gritsun (2020) which are based on the atmospheric stability during Atlantic blockings. Based on the $Chi^2$-skewness, we estimate a reduction of active degrees of freedom during Eurasian heat waves of about 25% compared to climatology.

3. **Intramonthly variance and entropy during heat waves:** We showed that intramonthly variance and entropy decrease at planetary scales (k=3) and increase at synoptic scales k=7-8, consistent with the well-known prevalence of atmospheric blocking regime during surface heat waves. However, for the limited dataset of heat waves, only the synoptic-scale variance and entropy increase is found statistically

significant. Future studies with longer datasets, such as climate model outputs are an opportunity for both models' validation and larger datasets of extreme events.

**Specific comments**

**Comment:** There is one sentence in the manuscript that I find confusing in Section 3.2: "Under the global variability spectrum, we imply the PDFs of the global energy anomalies and under signatures of heat waves, we imply significant changes in the distribution of energy anomalies." If the authors have new findings and plan to greatly revise and resubmit the manuscript, please rewrite this sentence with clarity.

**Response**: The sentence is rewritten. Its revised version explains that the term global variability spectrum refers to the PDFs of the normalised anomalies in global energy, and the effects (or signatures) of heat waves imply significant changes in the distribution of energy anomalies during the Eurasean heat waves.

---

## Author Comment (AC2)

Paper: wcd-2022-23, entitled "Signatures of midlatitude heat waves in global Rossby wave spectra",

By Iana Strigunova, Richard Blender, Frank Lunkeit, and Nedjeljka Žagar

**Response to the comments by Referee RC2**

https://doi.org/10.5194/wcd-2022-23-RC2

Dear Referee,

Thank you very much for your constructive comments on the manuscript.

Following your comments and comments from another reviewer and the editor, we have been largely rewriting the manuscript in an effort to highlight the original aspects of our method and the originality and added value of our results. While the results of statistics remain unaltered, we plan to extend the analysis by showing the zonal- and meridional-scale dependent entropy reduction during the Eurasian heat waves in relation to intramonthly variance reduction. For this purpose, the results section is being extended and some new figures will be added. We also plan to replace 'midlatitude' by 'Eurasian' in the title, as a more correct wording for the paper content.

Enclosed please find our responses, presented in blue font following your comments in black font.

Your sincerely,

Iana Strigunova, Richard Blender, Frank Lunkeit, and Nedjeljka Žagar

**Comment**
The authors have applied three-dimensional normal mode decomposition to wind and geopotential fields to investigate structural differences of European heat waves in modal space relative to climatology. They find the skewness of PDFS of planetary-scale circulation is increased by a factor of two, and variance decreases for planetary scales and increases for synoptic scales during the heat waves. Overall, I find this study can provide a unique perspective of heat wave characteristics in modal space, but they may need to put more efforts into interpreting and presenting the results. Below I list several concerns:
1. Please address the significance of the difference in Figs.6,8. Because there are limited samples for the heat waves, is it possible that the diffidence is caused by sampling?

**Response**: Thank you very much for your comment. We have added in Fig. 8 (to become Figure 8a) the 95%-confidence intervals of the results as obtained through bootstrapping with 1000 simulations. Preliminary figure is included showing that the intramonthly variance

reduction in zonal wavenumbers 7-8 is statistically significant whereas the intramonthly variance increase in the zonal wavenumber 3 is not. The analysis will be complemented by discussing the associated changes in the zonal mean state, not shown in Figure 8.

[Figure]

Figure 1: Time-averaged intramonthly variance spectra of Rossby waves for climatology (blue) and Eurasian heat waves (magenta). Averaging is performed over a 40-year period 1980-2019, months May-Sep of ERA5. The embedded bottom left panel shows percentage of relative change whereas the top right inset displays zoomed spectra for k=6-9. The 95%-confidence intervals (blue shading) are obtained through bootstrapping with replacement with 1000 simulations.

**Comment:** Based on the presented results, one may also get the impression that heat waves are structurally similar to climatology, except that the amplitude is higher. Should we emphasize the similarity or the difference?

**Response**: We emphasize differences and add information associated with our holistic method and related results while we also provide references to recent studies of extremes. The revised paper will thus include both aspects and be significantly re-written. For example, we cite previous research by Galfi and Lucarini (2021) of surface heat waves using Large Deviation Theory and by Lucarini and Gritsun (2020) of blockings as manifestations of Unstable Periodic Orbits. They found that the persistent atmospheric patterns associated with surface heat waves are not typical (in the statistical sense) compared to the climatology, but follow a dynamics which is already encoded in the natural climate variability.

**Comment**: Why are the amplitudes of the two PDFs so similar in Fig.6, while the total number of heatwaves is much smaller than the number of cases used to calculate climatology? I am not sure whether I understand how they defined climatology.

**Response**: The amplitudes are comparable due to the normalization. The normalization is necessary in order to account for the red energy spectrum with largely different amplitudes associated with various vertical modes. This can be seen from the equation for the computation of modal energy that shows that energy in a single mode involves a multiplication by the equivalent depth. The differences between the climatology and Eurasian heat waves are illustrated in the enclosed figure that shows empirical distribution functions (ECDFs) of normalized total energy anomalies (a) all zonal wavenumbers k, (b) the zonal mean state (k = 0), (c) planetary-scale waves (k=1-3), (d) synoptic-scale waves (k=4-10). Reanalysis dataset

distributions are depicted as green dots for extended boreal summer (MJJAS) 1980-2014 (1980-2019 for ERA5). The distributions during heat waves (denoted "only heat waves") are depicted as red dots with the values of skewness indicated in the panels.

[Figure]

**Comment:** What's the reason to normalize energy anomalies? Does it impact the major results?

**Response**: The normalization is necessary in order to account for the red energy spectrum with largely different amplitudes associated with various vertical modes. This can be seen from the equation for the computation of modal energy that shows that energy in a single mode involves a multiplication by the equivalent depth. The normalization thus allows us to easily perform statistics in various vertical modes, and to combine visually otherwise different PDFs. We shall explain this aspect in more detail in the revised paper.

---

## Author Response (AR1)

**Paper: wcd-2022-23, entitled "Signatures of midlatitude heat waves in global Rossby wave spectra",**

By Iana Strigunova, Richard Blender, Frank Lunkeit, and Nedjeljka Žagar

Dear Dr. Rivière,

Thank you very much for your thorough and constructive comments on the manuscript.

Following your comments and comments from the two reviewers, we have largely rewritten the manuscript in an effort to highlight the original aspects of our method and the originality and added value of our results. We have extended the literature overview and added further Results. While the results of statistics remain unaltered, the extended analysis shows the zonally and meridionally scale-dependent entropy reduction during the Eurasian heat waves in relation to intramonthly variance reduction. For this purpose, the extended Results section includes two new figures (new Fig. 8 and Fig. 9). We have changed the title of the manuscript; 'midlatitude' has been replaced by 'Eurasian' to better reflect the focus of the paper. Due to extensive changes in the paper, we do not list the line numbers where the changes took places or provide exact citations of the new lines.

Enclosed please find our responses, presented in blue font following your comments in black font. We are looking forward to hearing from you again.

Your sincerely,

Iana Strigunova, Richard Blender, Frank Lunkeit, and Nedjeljka Žagar

**Major comments:**

**Comment**

1) Originality of the results and more in depth analysis. The authors consider a new approach to tackle the problem of the link between Rossby wave circulation and heat waves over Eurasia, which is based on projecting the data onto normal mode functions. According to the authors, the main advantage compared to one-dimensional Fourier method is to keep horizontal and vertical coherency of the modes. While the reader could be sensitive to this argument, the results themselves do not

bring new information on that aspect. For instance, we would be interested to know what are the main modes excited during heat waves. So far the information gained from the paper is quite succinct and is already documented in other papers: heat waves are favored by planetary-scale Rossby waves that have mainly a barotropic structure and project mainly onto blocking. Also we learn that the zonal mean flow is weaker than usual for those events as also expected and described in other studies. One key result provided by the conclusion is that the number of degrees of freedom forming the planetary wave energy anomalies of these events is reduced. Could we translate it in saying that the number of modes excited during heat waves is reduced ? If yes, what are their shape ? We understand it is mainly zonal wavenumber 2, 3 but since the method provide meridional and vertical structures of these modes, the reader would like to know more about those structures. At least such an information could be one key result that can only be provided by the adopted approach.

**Response:** We regret that the originality of the methodology and results was not clearly stated in the previous version of the manuscript. In addition, we may not have emphasized enough the main focus of our study, which is on the signature of Eurasian heat waves in the statistics of the global circulation rather than on particular physical processes. We also link our findings to previous studies on the phenomenology and mechanisms of Eurasian heat waves to interpret our results and evaluate their relevance. The largely re-written paper and new elements of the Results section make this hopefully more clear. We tried to make the goals of our study clear in the new introduction.

We find that during heat waves, a change in the skewness of the probability density functions (PDFs) of the zonal mean state, planetary- and synoptic-scale Rossby waves, although the results obtained for the zonal mean state and synoptic-scale Rossby waves are found to be insignificant. An increase in skewness hints at a decrease in the active degrees of freedom. This result may indicate that less planetary-scale and more synoptic-scale Rossby modes (rather than the exact number of reduced or excited single modes) independently contribute to variability during heat waves. In the revised version, we demonstrated that intramonthly variance and entropy decrease at planetary scales and increase at synoptic scales. However, although the reduction of the active degrees of freedom is an important feature, explicit identification of the involved modes and the mechanism leading to the reduction is hardly feasible due to the complex dynamics. We note this with a clearer definition of our use of active degrees of freedom in the new introduction.

**Comment**

2) Clarity of writing.

a) Abstract: it should be self explanatory. As it is, it is hard to extract information. First, the mention of the method is not clear enough to me. Line 2: the sentence "... circulation is defined in terms of Rossby wave solutions of the linearized primitive equations" is not self explanatory. First the use of reanalysis datasets should be mentioned. Then I would expect a more precise sentence of this type: "each variable from reanalysis is written as the sum of projections onto global normal mode functions that are themselves obtained from a linearization of the primitive equations with an an atmosphere at rest. The normal-mode function decomposition is performed in wavenumber space defined by the zonal wavenumbers, meridional modes and vertical structure functions. The circulation variability .... climatological values." Also line 7, number of degrees of freedom is a bit obscure and since it is related to the main result of the paper this should be well explicited.

**Response:** We have revised the abstract by adding information about the reanalysis datasets, reducing the presentation of the method' to mentioning Hough expansion coefficients representing Rossby modes, and focusing on the main findings. The discussion of the degrees of freedom is no longer a part of the abstract due to the focus on more relevant, new results, but we as discuss it in more detail in the Conclusions as "a hint" from obtained skewness results.

**Comment**

b) Introduction:

- The first paragraph recalling the main results on the relationship between Rossby waves and heat waves is quite short. The number of citations is rather limited and the paragraph does not provide a global overview of the current literature. For instance, the studies by Tang and Branstator should be recalled in that paragraph. Also the description of each individual studies is rather unprecise: the notion of persistence, quasi-stationary waves is not mentioned.

Moreover, results of previous studies (e.g., Zagar et al., 2018, 2020) that have used the NMF decomposition are not summarized. To which purpose/subject has such a method been applied so far ? It seems that Zagar et al (2020) already found important results on summertime Rossby waves that are not summarized. Lines 34-35: these lines appear quite suddenly at the end of the paragraph and no smooth transition to the NMF decomposition is provided. Please also mention why the NMD method is expected to work well in midlatitudes. In Zagar et al (2020) it is said to be more adapted to the tropics.

- The 3rd paragraph is very difficult to understand without having read the method section. Besides, I am not sure the information is really important. If the authors think it is important, this should be more accurately explained.

- The 4th paragraph: same as the 3rd one. The sentences provide some mathematical properties without explicit implication for the atmosphere.

- The introduction does not mention the region of interest (Eurasia) whereas there are many case studies detailing mechanisms of heat waves in that region that should be cited.

So the introduction should more clearly and more smoothly introduce the method and should state the advantages of such a method. Also the exact objectives of the paper are not provided. Is it just to test a new decomposition to study Rossby waves and their links with heat waves ? I found the introduction of Zagar et al (2020) much clearer in that sense.

**Response:** We have largely revised the introduction to make clear our specific goals, method and differences from earlier studies of heat waves and studies using normal-mode function decomposition. As changes are many, here we just summarise them:

- We have extended the introductory paragraph by providing a better overview of the research of heat waves and blocking. However, we do not go into many details about the drivers, onset and dynamics of blocking events or Eurasian heat waves as this is not a process study.
- Goals are stated more clear in order to clarify that we perform statistical diagnosis of heat waves in modal space, and we do not study the dyanmics of these events.
- We refer to the relevant findings of Žagar et al. (2020) and clarify that they looked at global intraseasonal scales including the tropical variability because they used all vertical modes,

also baroclinic vertical modes. They also did not attempt to verify whether the trends found in the Rossby waves reflect the midlatitude heat waves.

- The third and fourth paragraphs are merged and revised to keep the information required to understand the paper's findings. In particular, why the normalized energy anomalies are skewed, how skewness and excess kurtosis are connected with degrees of freedom and how the number of degrees of freedom are associated with the number of single modes.
- The method introduction has been smoothed. We have rephrased the description of our approach from "more rigorous" to "holistic" pointing out the completeness rather than possible advantage compared to the 1D Fourier series approach.

**Comment**

c) Method: I found some lack of rigor in the presentation of the method. Here also the presentation of the method in Zagar et al. (2020) is clearer. Of course the aim is not to repeat the same description here but the reader expects some coherency and would like to have all the notations properly introduced; it is not the case for u,v, h (geopotential ?), sigma (terrain following vertical level ?). The fact the modes are obtained by a linearization of the primitive equations with an atmosphere at rest is not mentioned. Maybe introducing the time coordinate would be good especially because we will look at them in submonthly variance spectra.

**Response:** We have revised the description of the method and we begin by introducing notations for winds (u, v), geopotential height (h) and describing sigma as "the ratio of the vertical level pressure and the surface pressure". Although we mention in the Conclusions that structures of the Rossby modes used for the projection are those of eigensolutions of the linearized primitive equations on the sphere with a basic state at rest, this is really not crucial for the results as we do not consider propagation properties (affected by the background state) in the projection. Due to the NMF method being stronger established as a diagnostic tool only more recently, this argument has been raised and repeatedly discussed in earlier papers n resonse to reviewers' questions.

The rest of the Method and Data section is carefully revised so that the appraoch is made clear. It is explained that the time coordinate is defined by time step (daily for the energy and its anomalies and monthly for intramonthly variance spectra).

**Comment:** d) Conclusion: The paper confirms lots of studies on the link between planetary-scale waves, mean flow, and heat waves but no references are provided in the conclusion

**Response:** In the revised Conclusion, we have clarified our findings, we have provided their quantification, and we added several relevant references. We have also added reference to Wirth and Polster (2021, J.Atmos.Sci.) to point to another perspective on the relation between Rossby waves and double jet formation.

**Comment:** 3) Usefulness of 4 reanalysis datasets. It is not clear to me why ERA-interim JRA-55 and MERRA are useful in that study since they bring only 13 heat waves events. Is there a figure showing all reanalysis datasets ?

**Response:** We have added in the revised version, around line 163 bottom of page 7): "Although only 13 heat waves are identified in ERA-Interim, JRA-55 and MERRA due to the shorter datasets, we include them to increase the sample size. We note that all 13 heat waves are identified in all reanalyses, but the individual datasets differ, as can be inferred from the number of detected days. Thus, we consider them as different realizations." See also Fig. 1 below for a comparison of the planetary-scale Rossby wave PDFs during heat waves.

[Figure]

Figure 1. The normalized energy anomalies distribution of planetary-scale Rossby waves (k=1-3, n=1-49, m=1-5) only during heat waves. The mean (all reanalyses combined) is depicted as a dashed black line. The solid yellow line depicts ERA5. The same for ERA-Interim (green), JRA-55 (blue), and MERRA (magenta).

**Comment:** 4) Title: The title should mention the region of interest since the focus is on heat waves in Eurasian region and not all the midlatitudes.

**Response:** The title has been changed accordingly.

**Minor comments:**

**Comment:** 1) Line 7: "number of degrees of freedom" is not clear enough. Can we replace it by "number of implied normal modes" ?

**Response:** We have added clarification in line 47: **"**In our analysis, the number of degrees of freedom is the number of all possible modes used in the projection, while the number of active degrees of freedom is a measure of the concentration of energy in large wavenumbers during a heat wave". More generally, active degrees of freedom is a qualitative measure of independent modes contributing to the dynamics or its complexity. We have added a note on our interpretation of active in contrast to degrees of freedom in the introduction.

**Comment:** 2) Line 29: please characterize more precisely the notion of quasi-resonance.

**Response:** In the revised version, we have introduced the notion of quasi-resonance more precisely: "amplified quasi-stationary Rossby waves due to the resonance with free waves trapped within the waveguide".

**Comment:** 3) Line 33: please detail the acronym "NMF"

**Response:** The acronym "NMF" has been expanded.

**Comment:** 4) Line 43: I am not sure "Rossby circulation" makes senses to everyone. Could it be replaced by: "to estimate the distribution of Rossby waves and their total energy."?

**Response:** We have replaced "Rossby circulation" with "scale-dependent PDFs of the Rossby waves and the zonal mean state".

**Comment:** 5) Line 72: please introduce u, v and h here.

**Response:** The variables have been introduced.

**Comment:**  6) Line 87: it is not clear that modes are extracted from a linearization of the primitive equations with an atmosphere at rest.

**Response:** In the revised version, we have specified that the 3D data is projected "on the Rossby and inertia-gravity eigensolutions of the linearized primitive equations on the sphere with a basic state at rest".

**Comment:** 7) Line 122: What is the reference for the MODES software ?

**Response:** The reference for the MODES software is https://modes.cen.uni-hamburg.de. The explanation of software is in Žagar et al., 2015. We have added a footnote with the link and cited the paper. (lines N)

**Reference**

Žagar, N., Kasahara, A., Terasaki, K., Tribbia, J., and Tanaka, H.: Normal-mode function representation of global 3D datasets: open-access software for the atmospheric research community, Geosci. Model Dev., 8, 1169–1195, https://doi.org/https://doi.org/10.5194/gmd-8-1169-2015, 2015.

**Comment:** 8) Line 132: I do not understand the choice of the study area.

**Response:** We have revised the sentence by pointing out the influence of heat waves on the chosen area.

**Comment:** 9) Line 147: what is meant by "significant part of the mid-latitude circulation" ?

**Response:** Under "significant part of the mid-latitude circulation" we meant the large part of variability in midlatitudes. We have modified the text accordingly.

**Comment:** 10) Line 176: It is not clear how the 4 datasets are used to create Fig. 5.

**Response:** We have revised as:"The skewness of the PDFs in Fig. 6 is based on the four reanalyses (ERA5, ERA-I, JRA-55, MERRA) combined. We apply bootstrapping with a replacement for different wavenumber ranges for a more robust statistical analysis (Fig. 5). Note that all values from reanalysis datasets are within the defined 95% confidence intervals (CI) for each wavenumber range (here is not shown)."

**Comment:** 11) Line 182: If "degrees of freedom" means "number of single modes" this should be more clearly stated before in the paper.

**Response:** We have clarified it in the revised introduction.

**Comment:** 12) Line 193-194: Where is the kurtosis computation shown in Fig. 5 ?

**Response:** The kurtosis computation is now shown in line N.

**Comment:** 13) Line 197: I do not see the link with Coumou et al (2015)'s paper since they work on trends in eddy kinetic energy and mean flow and they both exhibit a reduction.

**Response:** We have removed the citation of Coumou et al. (2015) and referred to Comou et al. (2014) instead, as it shows the connection between amplified Rossby waves, weakened zonal flow and surface extremes more explicitly.

**Reference**

Coumou, D., Petoukhov, V., Rahmstorf, S., Petri, S., and Schellnhuber, H. J.: Quasi-resonant circulation regimes and hemispheric synchronization of extreme weather in boreal summer, Proc. Natl. Acad. Sci. U.S.A., 111, 12 331–12 336, https://doi.org/10.1073/pnas.1412797111, 2014.

**Comment:** 14) Line 202: it is not clear how df is defined. Which variables are you talking about ?

**Response:** "df" stands for "degrees of freedom", and it "is the number of squares of the independent Gaussian variables with a unit variance which defines the Chi^2-distributed variable.

**Comment:** 15) Line 216: where is it shown ?

**Response:** The paragraph aims to demonstrate that the results obtained for global circulation statistics are not affected by Rossby waves activity in the Southern Hemisphere. We have revised the paragraph for clarity, which can be found at the end of Section 3.2.

**Comment:** 16) Line 233: why is it said that "surface extremes modify atmospheric circulation"? It does not make sense to me.

**Response:** We have rephrased it as "how atmospheric circulation is changed during surface extremes ".

**Comment:** 17) Line 239: "chi_nu" is not introduced in the method section

**Response:** We have introduced this notation in the method section and made corresponding changes across the manuscript.

**Comment:** 18) Figure 8: Is the decrease in variance by 5% significant ?

**Response:** The decrease is found statistically insignificant. For the limited dataset of heat waves, only the synoptic-scale variance increase is found statistically significant.

**Comment:** 19) Line 257-258: please recall in which sense the decomposition method provides an advantage compared to univariate filtering.

**Response:** To be more clear, we have modified the sentence in the conclusion: "Scale-selective Rossby wave filtering in physical space is seen as an advantage compared to univariate filtering using the Fourier series along the latitude circles, as the selected modes represent the relevant 3D dynamical fields (in terms of u, v and h)."

**Comment:** 20) Line 267: please reference the studies that have initially shown that result.

**Response:** The sentence is removed in the revised version to highlight the findings on global statistics in modal space rather than to underline results from existing studies in physical space.

**Comment:** 21) Line 273-275: Is the result on submonthly variance be related to persistence characteristics ? Please be more precise in the results obtained in the present paper and their physical interpretation.

**Response:** We have included the references about increased variance in the synoptic scales is consistent with increased synoptic activity during blocking (Shutts, 1983; Yamazaki and Itoh, 2013)."

**References**

1. Shutts, G.: The propagation of eddies in diffluent jetstreams: Eddy vorticity forcing of 'blocking'flow fields, Q. J. R. Meteorol. Soc., 109, 737–761, https://doi.org/10.1002/qj.49710946204, 1983.

2. Yamazaki, A. and Itoh, H.: Vortex–vortex interactions for the maintenance of blocking. Part I: The selective absorption mechanism and a case study, J. Atmos. Sci., 70, 725–742, https://doi.org/10.1175/JAS-D-11-0295.1, 2013.

Paper: wcd-2022-23, entitled "Signatures of midlatitude heat waves in global Rossby wave spectra",

By Iana Strigunova, Richard Blender, Frank Lunkeit, and Nedjeljka Žagar

**Response to the comments by Referee RC1**

https://doi.org/10.5194/wcd-2022-23-RC1

Dear Referee,

Thank you very much for the comments on our manuscript.

Following your comments and comments from another reviewer and the editor, we have largely rewritten the manuscript in an effort to highlight the original aspects of our method and the originality and added value of our results. While the results of statistics remain unaltered, we have extended the analysis by showing the zonal- and meridional-scale dependent entropy reduction during the Eurasian heat waves in relation to intramonthly variance reduction. For this purpose, the results section has been extended and new figures (Fig. 8 and Fig. 9) have been added. We have also modified the title and have replaced 'midlatitude' by 'Eurasian' in the title, as a more correct wording for the paper content.

Enclosed please find our response, presented in blue font follow your comments in black font.

Your sincerely,

Iana Strigunova, Richard Blender, Frank Lunkeit, and Nedjeljka Žagar

**Comment:**
**Summary of the article**
This manuscript isolates heat waves events, defined as episodes which spatially averaged surface temperature in Eurasia are above 95% percentile on at least 3 consecutive days, and compare their spatial spectra of energy anomalies of various ranges of wavenumbers with the climatology. This work finds that during heat waves, the skewness of planetary waves grows while that of the zonal mean flow goes in opposite directions, which is consistent with previous findings.
**General comments**

I am concerned about the fact that the analyses done in this work all confirm previous findings but provide no additional insights to how we understand heat waves/blocking. It has been well-known that heat waves are associated with blocking since blocking, by definition, refers to the phenomena in which (1) the eastward zonal wind is disrupted (increase in wave amplitude essentially implies weakened zonal wind, as implied by non-acceleration theorem) (2) the high pressure cell remains stagnant for a few days, (Therefore, reversal of zonal wind and geopotential height anomalies have been used to define/detect blocking.) which, this manuscript essentially shows.

There have been many studies showing blocking being the culprit of heat waves in the eurasia region as confirmed by the analyses here. Therefore, in order to be accepted for publication, the authors should have shown what extra insights about heat waves can we obtain from these analyses, which I think the current version of manuscript is lacking.

**Response**:

We thank the Reviewer for the comments expressing a concern about the lack of added insights about heat waves in our analysis. We think that our efforts to validate the use of three-dimensional, global spectral decomposition in terms of normal-mode functions as a suitable tool for signatures of regional heat waves in global circulation, somehow masked the originality and added value of our results. We have rewritten parts of the paper to point out original aspects of our study. To our knowledge, previous studies did not quantify the skewness of global circulation at different scales of the Rossby waves during the Eurasian heat waves. We are neither aware of other research showing what aspects of the global circulation, as measured by PDFs of anomalies in the total energy, are affected by the Eurasian heat waves. Novelties of our study can be split in three parts:

1. **Method:** Our method is novel in its identification of Rossby waves using a multivariate spherical projection on the Rossby eigensolutions of the linearized primitive equations. This provides time series of Rossby waves in terms of Hough harmonics, that can be seen as the spherical equivalents of the geostrophic stream function on the midlatitude beta plane. Scale-selective Rossby wave filtering in physical space is seen as an advantage compared to univariate filtering using the Fourier series along the latitude circles. While we focus on Rossby waves or balanced dynamics, the difference to the total signal in terms of inertia-gravity modes can be used to analyze whether midlatitude ageostrophic flow (unbalanced dynamics) increases during the Eurasian heat waves. We have made changes across the paper to make this clear.

2. **Statistical properties of global balanced circulation during Eurasian heat waves:** We combined the four modern reanalysis datasets to provide robust results. We show that the energy distribution of a single Rossby mode follows a Chi^2-distribution. Our scale-dependent statistics, performed on the normalized energy anomalies, shows that the energy distributions of the zonal mean state (zonal wavenumber k=0) and of the planetary-scale (k=1-3) circulation are more skewed than the distributions at synoptic and smaller scales, with extended right tails. During the Eurasian heat waves, the skewness in planetary waves increases while the opposite occurs in the zonal mean flow. The increase in skewness can be linked with a decrease in the number of active degrees of freedom in state space during heat waves. This aligns with the results of Lucarini and Gritsun (2020) which are based on the atmospheric stability during Atlantic blockings. Based on the Chi^2-skewness, we

estimate a reduction of active degrees of freedom during Eurasian heat waves of about 25% compared to climatology. We added this reference in the conclusion.

3. **Intramonthly variance and entropy during heat waves:** At the end of conclusion, we now write: "We furthermore showed that during the Eurasian heat waves intramonthly variance and entropy decrease at planetary scale (k=3) and increase in synoptic scales k=7-8. The reduction of intramonthly variance for planetary scales may be linked to a larger persistence of large-scale anomalies during blocking (e.g., Legras and Ghil, 1985). On the other hand, the increase of variance in the synoptic scales is consistent with increased synoptic activity during blocking (e.g., Shutts, 1983; Yamazaki and Itoh, 2013). We would like to note that for the limited dataset of heat waves, only the synoptic-scale variance and entropy increase is found to be statistically significant. Future studies with longer datasets, such as climate model outputs are an opportunity for both models' validation and larger datasets of extreme events.

**References**

1. Legras, B. and Ghil, M.: Persistent anomalies, blocking and variations in atmospheric predictability, J Atmos Sci, 42, 433–471, https://doi.org/10.1175/1520-0469(1985)042<0433:PABAVI>2.0.CO;2, 1985

2. Lucarini, V. and Gritsun, A.: A new mathematical framework for atmospheric blocking events, Clim. Dyn., 54, 575–598, https://doi.org/10.1007/s00382-019-05018-2, 2020.

3. Shutts, G.: The propagation of eddies in diffluent jetstreams: Eddy vorticity forcing of 'blocking'flow fields, Q. J. R. Meteorol. Soc., 109, 737–761, https://doi.org/10.1002/qj.49710946204, 1983.

4. Yamazaki, A. and Itoh, H.: Vortex–vortex interactions for the maintenance of blocking. Part I: The selective absorption mechanism and a case study, J. Atmos. Sci., 70, 725–742, https://doi.org/10.1175/JAS-D-11-0295.1, 2013.

**Specific comments**

**Comment:** There is one sentence in the manuscript that I find confusing in Section 3.2: "Under the global variability spectrum, we imply the PDFs of the global energy anomalies and under signatures of heat waves, we imply significant changes in the distribution of energy anomalies." If the authors have new findings and plan to greatly revise and resubmit the manuscript, please rewrite this sentence with clarity.

**Response**: The sentence is rewritten. "Here, the term global variability spectrum refers to the PDFs of the normalized anomalies in global energy, and the effects (or signatures) of heat waves implies significant changes in the distribution of energy anomalies."

Paper: wcd-2022-23, entitled "Signatures of midlatitude heat waves in global Rossby wave spectra",

By Iana Strigunova, Richard Blender, Frank Lunkeit, and Nedjeljka Žagar

**Response to the comments by Referee RC2**

https://doi.org/10.5194/wcd-2022-23-RC2

Dear Referee,

Thank you very much for your constructive comments on the manuscript.

Following your comments and comments from another reviewer and the editor, we have been largely rewriting the manuscript in an effort to highlight the original aspects of our method and the originality and added value of our results. While the results of statistics remain unaltered, we extended the analysis by showing the zonal- and meridional-scale dependent entropy reduction during the Eurasian heat waves in relation to intramonthly variance reduction. For this purpose, the results section is extended and some new figures are added. We also replaced 'midlatitude' by 'Eurasian' in the title, as a more correct wording for the paper content.

Enclosed please find our responses, presented in blue font following your comments in black font.

Your sincerely,

Iana Strigunova, Richard Blender, Frank Lunkeit, and Nedjeljka Žagar

**Comment**
The authors have applied three-dimensional normal mode decomposition to wind and geopotential fields to investigate structural differences of European heat waves in modal space relative to climatology. They find the skewness of PDFS of planetary-scale circulation is increased by a factor of two, and variance decreases for planetary scales and increases for synoptic scales during the heat waves. Overall, I find this study can provide a unique

perspective of heat wave characteristics in modal space, but they may need to put more efforts into interpreting and presenting the results. Below I list several concerns:

1. Please address the significance of the difference in Figs.6,8. Because there are limited samples for the heat waves, is it possible that the diffidence is caused by sampling?

**Response**: Thank you very much for your comment. We address significance of results in Figure 5 and 6 by applying bootstrapping to deal with limited sample size. We have added in the new Fig. 8a the 95%-confidence intervals of the results as obtained through bootstrapping with 1000 simulations. It shows that the intramonthly variance increase in zonal wavenumbers 7-8 is statistically significant whereas the intramonthly variance decrease in the zonal wavenumber 3 is not. The analysis is complemented by discussing the associated changes in the zonal mean state, now shown in the new Fig. 8.

**Comment:** Based on the presented results, one may also get the impression that heat waves are structurally similar to climatology, except that the amplitude is higher. Should we emphasize the similarity or the difference?

**Response**: We emphasized differences and added information associated with our holistic method and related results while we also provided references to recent studies of heat waves in the revised introduction and conclusion. We refer to previous research by Galfi and Lucarini (2021) of surface heat waves using Large Deviation Theory and by Lucarini and Gritsun (2020) of blockings as manifestations of Unstable Periodic Orbits. They found that the persistent atmospheric patterns associated with surface heat waves are not typical (in the statistical sense) compared to the climatology, but follow a dynamics which is already encoded in the natural climate variability.

**References**

1. Galfi, V. M. and Lucarini, V.: Fingerprinting Heatwaves and Cold Spells and Assessing Their Response to Climate Change Using Large Deviation Theory, Phys. Rev. Lett., 127, 058 701, https://doi.org/10.1103/PhysRevLett.127.058701, 2021.

2. Lucarini, V. and Gritsun, A.: A new mathematical framework for atmospheric blocking events, Clim. Dyn., 54, 575–598, https://doi.org/10.1007/s00382-019-05018-2, 2020.

**Comment**: Why are the amplitudes of the two PDFs so similar in Fig.6, while the total number of heatwaves is much smaller than the number of cases used to calculate climatology? I am not sure whether I understand how they defined climatology.

**Response**: The amplitudes are comparable due to the normalization. The normalization is necessary in order to account for the red energy spectrum with largely different amplitudes associated with various vertical modes. This can be seen from the equation for the computation of modal energy that shows that energy in a single mode involves a multiplication by the equivalent depth. We added this information in the revised method

section. We also added the sentence: "The time series of all $I^{\prime}_{\nu}$ define the climatology".

**Comment:** What's the reason to normalize energy anomalies? Does it impact the major results?

**Response**: The normalization is necessary in order to account for the red energy spectrum with largely different amplitudes associated with various vertical modes. This can be seen from the equation for the computation of modal energy that shows that energy in a single mode involves a multiplication by the equivalent depth. The normalization thus allows us to easily perform statistics in various vertical modes, and to combine visually otherwise different PDFs. We explained this aspect in more detail in the revised paper in the method section.

---

## Author Response (AR2)

Paper: wcd-2022-23, entitled "Signatures of Eurasian heat waves in global Rossby wave spectra "

By Iana Strigunova, Richard Blender, Frank Lunkeit, and Nedjeljka Žagar

Dear Dr. Riviére,

Many thanks for your detailed, constructive comments on the revised manuscript. We appreciate it a lot and hope that this second version is a further improvement.

While making changes to address your comment, we performed minor technical changes across figures to make them easier to read and compare (for example, x-axes on the panel of PDFs in Figures 4 and 6 have been made equally). We have re-written the second paragraph of the abstract and made changes throughout the Results section and the Conclusions, aiming at a better presentation of the key findings.

Enclosed please find our responses, presented in blue font following your comments in black font. We are looking forward to hearing from you again.

Your sincerely,

Iana Strigunova, Richard Blender, Frank Lunkeit, and Nedjeljka Žagar

**Major comments:**

**Comment:**
1) abstract: The abstract is quite short and I am not sure it provides enough physical understanding. In particular, the second paragraph of the abstract is not easy to understand.

   a) The physical meaning of "the skewness in planetary waves" should be more clearly explained. There is space to describe the relationship between skewness and degrees of freedom.
   b) I would replace "the skewness in planetary waves" by " the skewness in planetary-wave energy".
   c) Similar to reviewer 1's comment, I found a bit strange to summarize more the results of section 3.4 and less those of sections 3.2 and 3.3. In my opinion, Fig.6d is an important figure showing the increased synoptic wave energy. Fig.8a completes Fig6d by showing this is mainly coming from wavenumbers 7,8.
   d) Maybe one way to describe the results is to start with the zonal mean flow, then planetary waves and finally finish with the synoptic wawes. As it is written now, everything is mixed up in this paragraph and makes the reader confused.

**Response:** We have re-written the second paragraph of the abstract addressing your comments. We (a) described the relationship between skewness and degrees of freedom, (b) referred to "energy", (c) aimed at a more complete synthesis of our results using (d) the suggested structure.

**Comment:**

2) Fig.8a and related discussion. The text stays largely focused on what we see on the figures. But there are not enough statements providing a physical meaning of all the quantities computed (see below for more detailed comments on that aspect).

Response: First, we removed Figure 8c and the entropy discussion from the paper completely. At this stage, we think that the entropy did not provide added information on the top of temporal variance.

**Comment:**

3) Conclusion: there is no attempt to provide a global picture. For instance, the two results on planetary waves are an increase in the skewness and a decrease in the intraseasonal variance. Can these two results be connected to provide a broad picture of the planetary wave behavior during heat waves? Or are these two results entirely independent?

**Response:** At the end of the conclusions, we added a new paragraph providing a broader and more physical picture of the circulation changes during heat waves based on our results. Revised discussion of Figure 8a provide references to Figure 6. This is reflected also in changes in the Conclusion and Abstract sections.

**Minor comments:**

**Comment:** Line 218: The sentence "We apply bootstrapping with a replacement for different wavenumber ranges for a more robust statistical analysis (Fig. 5)." is not clear to me.

**Response:** We modified for clarification as follows: "The robustness of the statistical analysis in Fig. 5 is checked by applying bootstrapping with replacement for skewness and excess kurtosis with 1000 realizations for every presented wavenumber range."

**Comment:** Figure 5 / caption: "replacement with 1000 simulations" is not clear to me. I am not sure "simulations" is the appropriate wording.

**Response:** We removed it from the caption as it is explained in the text and write that "Vertical lines mark 95%-confidence intervals.".

**Comment:** line 233-234: Fig5a--> Fig6a and Fig5b --> Fig6b

**Response:** We reformulated section 3.2.1 (the discussion of Figures 5 and 6). We now make clear which Figure we actually discuss, which was indeed misleading.

**Comments:**

- Paragraph from lines 233-240: Fig6d is an important figure and should be described.
- Lines 249-253: The discussion on the change in PDFs of synoptic Rossby waves is not precise enough and does not refer to any figure. In my opinion, Fig. 6d shows there is more energy is synoptic waves during heat waves as said in the text. But why is this figure not cited in the paragraph ?

**Response:** We reformulated section 3.2.1 (the discussion of Figures 5 and 6) to give a more thorough description and discussion of Figure 5 and Figure 6 with respect to the identified changes. As the changes are numerous, we do not copy them here but refer to the pdf of the revised paper with all changes highlighted.

**Comment:** Line 275-278: In my opinion, Fig7f confirms the amplification of wave-3 pattern during heat waves as shown in the cited studies. I think the authors should explicitly say this is a confirmation of previous studies (maybe in another geographical area ?).

**Response:** We now explicitly state that " The results in Fig.7 align with Teng and Branstator (2012) and Ragone and Bouchet (2021), where the zonal wavenumber k=3 pattern was found dominant for HWs that occurred in the US, France and Scandinavia. Therefore, the results demonstrate that changes in atmospheric circulation during surface extremes occur not only regionally but also in remote regions, similar to the idea of teleconnection patterns noted in recent studies (e.g. Kornhuber et al. 2019).

**Comment:** Line 286 / Eq (8): Is Chi_nu the same as Chi^k_n(m) ?

**Response:** Yes, this is defined in the 4th line of Section 2.3.

**Comment:** Eq. (9): what does the superscript "h" mean? I imagine it refers to heat waves. What about V_nu ? Is it a climatological value? or is it a value that is computed for every month?

**Response:** We added the clarification: "Intramonthly variance is computed for all months and averaged to create the climatological variance spectrum, $V_{\nu}$. The averaging over all months with heat waves produced the variance spectrum $V^h_{\nu}$ (Here we drop extra signs for the averaging operator)."

**Comment:** The physical meaning of the decreased variance in wavenumber 3 is never provided. Does it mean that wavenumber 3 is quasi-stationary during months marked by heat waves?

**Response:** We have expanded the discussion of Fig 8 and added respective changes in the Conclusions.

**Comment:** The zooms in Fig.8a are not clear enough to conclude about the significances of the reduction in wavenumber 3 variance and the increase in wavenumbers 7,8 variance. Both cases show values very close to the 95% confidence.

**Response:** We added the 95%-CI also in the inset figure of Figure 8a. The fact that the values are not far inside/outside the confidence interval is now clear.

**Comment:** The paragraph from lines 313 to 325 is not easy to read. Please insert "to" after "close" in line 317. I am not sure "barotropicity" exists. Maybe replace by "barotropic structure". The sentence starting by "Note that Fig.9" is difficult to understand.

**Response:** We have re-written and simplified this paragraph.

**Comment:**
Line 347-348: The authors should help the reader to physically interpret the skewness in energy anomalies. If the distributions of the zonal mean state and planetary scale circulation are more skewed than that of synoptic scales, what does that mean physically? Does it mean that the planetary wave energy is concentrated in a very limited number of planetary scale modes? In my mind, the reduction of the active degrees of freedom during heat waves means that there are only very specific modes that are excited. Am I right?

**Response:** We tried to give a physical interpretation of the skewness during the discussion of individual results and at the end of the conclusion by adding a "broad picture". The numerous changes can be seen in the annotated pdf.

Paper: wcd-2022-23, entitled "Signatures of Eurasian heat waves in global Rossby wave spectra "

By Iana Strigunova, Richard Blender, Frank Lunkeit, and Nedjeljka Žagar

**Response to the comments by Referee RC1 (second revision)**

https://doi.org/10.5194/wcd-2022-23-RC1

Dear Referee,

Thank you very much for the constructive comments on our manuscript.
Following your comments and comments from the editor, we have performed the additional revision of the paper taking chance to also improve some of figures technically. We have re-written the second paragraph of the abstract and made changes throughout the Results section and the Conclusions, aiming at better presentation of the key findings.

Enclosed please find our response, presented in blue font following your comments in black font.

Your sincerely,

Iana Strigunova, Richard Blender, Frank Lunkeit, and Nedjeljka Žagar

**Comment:**
The statement claiming that there is significant difference between intramonthly variance climatology and heat wave events is a bit weak given there were only a few heat waves events in the analysis to bootstrap from. (It is hard to read from Fig 8(a) that the percentage of relative change for k=6-9 has points outside the 95% confidence interval). This claim shall be verified with GCM simulations, which will be out of scope of the current study. It would be better to change the abstract a bit to focus on Results in session 3.1-3.3 and put less focus on 3.4.

**Response**:
As the Abstract has been largely rewritten, we do not reproduce it here. The revised second paragraph of the abstract hopefully offers a clearer presentation of all our key results.
Figure 8a has been improved by adding the confidence interval in the inset figure. We added the sentences about the sample size and the possible application of GCM simulations to gain more robust results. However, we also note that we do not expect large deviations to be found due to the global scale of analysed data.

**Specific comments**

**Comment:** I think the logic between the two sentences in lines 250-253 is not clear. Probably the authors have to add one more sentence before "More intensive cyclones and anticyclones

are found to maintain blocking…" to explain why their findings are consistent with the world of Shutts 1983 and Yamazaki and Itoh 2013.

**Response**: We have re-written the discussion of Figures 5 and 6 in section 3.2.1. New lines corresponding to the previous lines 25-253 state that "The shift can be interpreted as increased positive deviations in the synoptic-scale energy during heat waves. More energy in synoptic-scale circulation can be viewed as more intensive cyclones and anticyclones which are found to maintain blocking by eddy straining (Shutts, 1983) and selective absorption (Yamazaki and Itoh, 2013) mechanisms."

**Comment:** Fig. 8(b) and (c) would better be changed to "variance/entropy change" instead of "variance/entropy reduction", and have the color axis swapped such that red refers to increase and blue refers to reduction. Confusing double negative statements can be avoided.

**Response**: We removed Figure 8c and the entropy discussion from the paper completely. At this stage, we think that the entropy did not provide added information on the top of temporal variance. The figure title is changed to 'Variance change" as suggested and the color axis has been swapped.

---

## Author Response (AR3)

**Paper: wcd-2022-23, entitled "Signatures of midlatitude heat waves in global Rossby wave spectra",**

By Iana Strigunova, Richard Blender, Frank Lunkeit, and Nedjeljka Žagar

Dear Dr. Rivière,

Thank you very much for your additional thorough and constructive comments on the manuscript.

We have reviewed the manuscript accordingly, including a few more minor changes.

Enclosed please find our responses, presented in blue font following your comments in black font. We are looking forward to hearing from you again.

Your sincerely,

Iana Strigunova, Richard Blender, Frank Lunkeit, and Nedjeljka Žagar

**Comments & Responses:**

**Comment:**
1) Line 197: The time series of I'_nu is said to define the climatology. This is misleading for the reader since the climatology of the energy is in bracket in Eq(6). I would say I'_nu is the time series of the anomalous daily energies.

**Response:** We clarified that in the further analysis, the time series of the anomalous daily energies, I'_nu, is considered to be the climatological state (climatology) as a reference state for the comparison with the time series of the anomalous energies during heat waves. The latter is formed combining only time steps of the observed HWs according to Table 1.

**Comment:**
2) Line 216: I do not understand the if in parenthesis

**Response:** We removed "(if)"

**Comment:**
3) Lines 228-234: There is no introductory sentences to Fig. 5 and its objectives. The paragraph starts discussing robustness of the results and not the results themselves. This is not confortable for the reader.

**Response:** We added an introductory sentence to Fig. 5 and clarify that Fig. 5 is used here for discussing the climatology and, in the following section, the HWs.

**Comment:**

4) Section 3.2.1: There is an issue on the order of the figures and their description. I do not understand why Fig.6 appears after Fig.5 since the main results of Fig.5 are described after those of Fig.6.

**Response:** With reference to the previous comment, we now clarify that Fig. 5 is first used for the discussion of the climatology (in section 3.2.1), and the changes of the PDDFs due to HWs are discussed in the following section together with PDFs of HWs (Fig. 6).

**Comment:**

5) Line 312: "is also consistent"

**Response:** Corrected.

**Comment:**

6) Fig. 8b. Please indicate we are looking at variance anomalies of HWs with respect to climatology.

**Response:** We modified the caption of Fig. 8 accordingly.

**Comment:**

7) Lines 363-367. The sentences miss some precision. Suggestion: "The increase in skewness FOR PLANETARY WAVES REVEALS the decrease in the number of active degrees of freedom during HWs. This aligns with the results of Lucarini and Gritsun (2020) which are based on the atmospheric stability during Atlantic blockings. Based on the $\chi^2$ -skewness, we estimate a reduction of the active degrees of freedom FOR PLANETARY WAVES during Eurasian HWs of about 25% compared to climatology.

**Response:** Modified as suggested.

**Comment:**

8) Line 374: I do not understand the beginning of the sentence: "During HWs, planetary Rossby waves are less active (especially k = 3), and a persistent anomaly... ". Even though the intraseasonal variance of planetary wave k=3 decreases, this wavenumber emerges during HWs and is persistent. So I think saying "planetary Rossby waves are less active" is really confusing.

**Response:** We modified the text as follows: "During HWs, the planetary-scale Rossby waves (primarily k=3) exhibit reduced intramonthly variability. The involved modes are less independent from one another, and a persistent large-scale anomaly is formed, typically referred to as blocking. "